# Bellman Eluder Dimension: New Rich Classes of RL Problems, and Sample-Efficient Algorithms

**Chi Jin**
Princeton University
chij@princeton.edu

**Qinghua Liu**
Princeton University
qinghual@princeton.edu

**Sobhan Miryoosefi**
Princeton University
miryoosefi@cs.princeton.edu

## Abstract

Finding the minimal structural assumptions that empower sample-efficient learning is one of the most important research directions in Reinforcement Learning (RL). This paper advances our understanding of this fundamental question by introducing a new complexity measure—Bellman Eluder (BE) dimension. We show that the family of RL problems of low BE dimension is remarkably rich, which subsumes a vast majority of existing tractable RL problems including but not limited to tabular MDPs, linear MDPs, reactive POMDPs, low Bellman rank problems as well as low Eluder dimension problems. This paper further designs a new optimization-based algorithm—GOLF, and reanalyzes a hypothesis elimination-based algorithm—OLIVE [proposed in 1]. We prove that both algorithms learn the near-optimal policies of low BE dimension problems in a number of samples that is polynomial in all relevant parameters, but independent of the size of state-action space. Our regret and sample complexity results match or improve the best existing results for several well-known subclasses of low BE dimension problems.

## 1 Introduction

Modern Reinforcement Learning (RL) commonly engages practical problems with an enormous number of states, where *function approximation* must be deployed to approximate the true value function using functions from a prespecified function class. Function approximation, especially based on deep neural networks, lies at the heart of the recent practical successes of RL in domains such as Atari [2], Go [3], robotics [4], and dialogue systems [5].

Despite its empirical success, RL with function approximation raises a new series of theoretical challenges when comparing to the classic tabular RL: (1) *generalization*, to generalize knowledge from the visited states to the unvisited states due to the enormous state space. (2) *limited expressiveness*, to handle the complicated issues where true value functions or intermediate steps computed in the algorithm can be functions outside the prespecified function class. (3) *exploration*, to address the tradeoff between exploration and exploitation when above challenges are present.

Consequently, most existing theoretical results on efficient RL with function approximation rely on relatively strong structural assumptions. For instance, many require that the MDP admits a linear approximation [6–8], or that the model is precisely Linear Quadratic Regulator (LQR) [9–11]. Most of these structural assumptions rarely hold in practical applications. This naturally leads to one of the most fundamental questions in RL.

35th Conference on Neural Information Processing Systems (NeurIPS 2021).

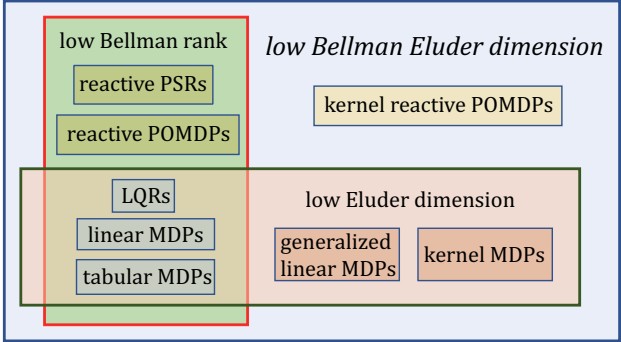

Figure 1: A schematic summarizing relations among families of RL problems[1]

**What are the minimal structural assumptions that empower sample-efficient RL?**

We advance our understanding of this grand question via the following two steps: (1) identify a rich class of RL problems (with weak structural assumptions) that cover many practical applications of interests; (2) design sample-efficient algorithms that provably learn any RL problem in this class.

The attempts to find weak or minimal structural assumptions that allow statistical learning can be traced in supervised learning where VC dimension [12] or Rademacher complexity [13] is proposed, or in online learning where Littlestone dimension [14] or sequential Rademacher complexity [15] is developed.

In the area of reinforcement learning, there are two intriguing lines of recent works that have made significant progress in this direction. To begin with, Jiang et al. [1] introduces a generic complexity notion—Bellman rank, which can be proved small for many RL problems including linear MDPs [7], reactive POMDPs [16], etc. [1] further propose an hypothesis elimination-based algorithm—OLIVE for sample-efficient learning of problems with low Bellman rank. On the other hand, recent work by Wang et al. [17] considers general function approximation with low Eluder dimension [18], and designs a UCB-style algorithm with regret guarantee. Noticeably, generalized linear MDPs [6] and kernel MDPs (see Appendix C) are subclasses of low Eluder dimension problems, but not low Bellman rank.

In this paper, we make the following three contributions.

- We introduce a new complexity measure for RL—Bellman Eluder (BE) dimension. We prove that the family of RL problems of low BE dimension is remarkably rich, which subsumes both low Bellman rank problems and low Eluder dimension problems—two arguably most generic tractable function classes so far in the literature (see Figure 1). The family of low BE dimension further includes new problems such as kernel reactive POMDPs (see Appendix C) which were not known to be sample-efficiently learnable.

- We design a new optimization-based algorithm—GOLF, which provably learns near-optimal policies of low BE dimension problems in a number of samples that is polynomial in all relevant parameters, but independent of the size of state-action space. Our regret or sample complexity guarantees match [8] which is minimax optimal when specified to the linear setting. Our rates further improve upon [1, 17] in low Bellman rank and low Eluder dimension settings, respectively.

- We reanalyze the hypothesis elimination based algorithm—OLIVE proposed in [1]. We show it can also learn RL problems with low BE dimension sample-efficiently, under slightly weaker assumptions but with worse sample complexity comparing to GOLF.

## 1.1 Related works

This section reviews prior theoretical works on RL, under Markov Decision Process (MDP) models.

---

[1]The family of low Bellman rank problems and low Bellman Eluder dimension problems include both Q-type and V-type variants. Please refer to Section 3.1 and Appendix B for more details.

We remark that there has been a long line of research on function approximation in the *batch RL* setting [see, e.g., 19–22]. In this setting, agents are provided with exploratory data or simulator, so that they do not need to explicitly address the challenge of exploration. In this paper, we do not make such assumption, and attack the exploration problem directly. In the following we focus exclusively on the RL results in the general setting where exploration is required.

**Tabular RL.** Tabular RL concerns MDPs with a small number of states and actions, which has been thoroughly studied in recent years [see, e.g., 23–30]. In the episodic setting with non-stationary dynamics, the best regret bound $\tilde{\mathcal{O}}(\sqrt{H^2|\mathcal{S}||\mathcal{A}|T})$ is achieved by both model-based [27] and model-free [30] algorithms. Moreover, the bound is proved to be minimax-optimal [29, 31]. This minimax bound suggests that when the state-action space is enormous, RL is information-theoretically hard without further structural assumptions.

**RL with linear function approximation.** A recent line of work studies RL with linear function approximation [see, e.g., 7, 6, 32, 8, 33–36] These papers assume certain completeness conditions, as well as the optimal value function can be well approximated by linear functions. Under one formulation of linear approximation, the minimax regret bound $\tilde{\mathcal{O}}(d\sqrt{T})$ is achieved by algorithm ELEANOR [8], where $d$ is the ambient dimension of the feature space.

**RL with general function approximation.** Beyond the linear setting, there is a flurry line of research studying RL with general function approximation [see, e.g., 37, 1, 36, 38, 17, 39, 40]. Among them, [1] and [17] are the closest to our work.

Jiang et al. [1] propose a complexity measure named Bellman rank and design an algorithm OLIVE with PAC guarantees for problems with low Bellman rank. We note that low Bellman rank is a special case of low BE dimension. When specialized to the low Bellman rank setting, our result for OLIVE exactly matches the guarantee in [1]. Our result for GOLF requires an additional completeness assumption, but provides sharper sample complexity guarantee.

Wang et al. [17] propose a UCB-type algorithm with a regret guarantee under the assumption that the function class has a low eluder dimension. Again, we will show that low Eluder dimension is a special case of low BE dimension. Comparing to [17], our algorithm GOLF works under a weaker completeness assumption, with a better regret guarantee.

Finally, we remark that the algorithms proposed in [1, 33, 41] and this paper are all computationally inefficient in general. We notice several existing works [e.g., 7, 17] can be computationally efficient given suitable regression oracles but they require stronger representation conditions and also achieve worse regret guarantees.

**Relation to bilinear classes** Concurrent to this work, Du et al. [41] propose a new general tractable class of RL problems—bilinear class with low effective dimension (also known as low critical information gain in Du et al. [41]). We comment on the similarities and differences between two works as follows.

In terms of algorithms, both Algorithm 2 in this paper and the algorithm proposed in Du et al. [41] are based on OLIVE originally proposed in Jiang et al. [1]. The two algorithms share similar guarantees in terms of assumptions and complexity results. More importantly, our work further develops a new type of algorithm for general function approximation—GOLF, a natural and clean algorithm which can be viewed as an optimistic version of classical algorithm—Fitted Q-Iteration [42]. GOLF gives much sharper sample complexity guarantees compared to [41] for various settings, and is minimax-optimal when applied to the linear setting [8].

In terms of richness of new classes identified, it depends on (a) what structure of MDP the complexity measures are applied to, and (b) what complexity measures are used. For (a), BE dimension applies to the Bellman error, while the bilinear class allows general surrogate losses of the Bellman error. For (b), this paper uses Eluder dimension while Du et al. [41] uses effective dimension. It can be shown that low effective dimension always implies low Eluder dimension (see Appendix C.2). In short, Du et al. [41] is more general in (a), while our work is more general in (b). As a result, neither work fully captures the other.

In particular, our BE framework covers a majority of the examples identified in Du et al. [41] including low occupancy complexity, linear $Q^\star/V^\star$, $Q^\star$ state aggregation, feature selection/FLAMBE.

Nevertheless, our work can not address examples with model-based function approximation (e.g., low witness rank [36]) while [41] can. On the other hand, Du et al. [41] can not address the class of RL problems with low Eluder dimension [17] while our work can. Moreover, for several classes of RL problems that both works cover, our complexity measure is sharper. For example, in the setting of function approximation with generalized linear functions, the BE dimension is $\tilde{O}(d)$ where $d$ is the ambient dimension of the feature vectors, while the effective dimension under the generalized bilinear framework of Du et al. [41] is at least $\tilde{\Omega}(d^2)$.

## 1.2 Paper organization

We present preliminaries in Section 2, the definition of Bellman-Eluder dimension as well as its relations to existing complexity notions in Section 3. We present the results for algorithm GOLF in Section 4, and conclude in Section 5. Due to space limit, we postpone the results for algorithm OLIVE to Appendix A. Further discussions on Q-type versus V-type variants of BE dimension, as well as the practical examples will be provided in Appendix B and C. All the proofs are postponed to the appendix.

## 2 Preliminaries

We consider episodic Markov Decision Process (MDP), denoted by $\mathcal{M} = (\mathcal{S}, \mathcal{A}, H, \mathbb{P}, r)$, where $\mathcal{S}$ is the state space, $\mathcal{A}$ is the action space, $H$ is the number of steps in each episode, $\mathbb{P} = \{\mathbb{P}_h\}_{h \in [H]}$ is the collection of transition measures with $\mathbb{P}_h(s' \mid s, a)$ equal to the probability of transiting to $s'$ after taking action $a$ at state $s$ at the $h^{\text{th}}$ step, and $r = \{r_h\}_{h \in [H]}$ is the collection of reward functions with $r_h(s, a)$ equal to the deterministic reward received after taking action $a$ at state $s$ at the $h^{\text{th}}$ step. [2] Throughout this paper, we assume reward is non-negative, and $\sum_{h=1}^{H} r_h(s_h, a_h) \leq 1$ for all possible sequence $(s_1, a_1, \ldots, s_H, a_H)$.

In each episode, the agent starts at a *fixed* initial state $s_1$. Then, at each step $h \in [H]$, the agent observes its current state $s_h$, takes action $a_h$, receives reward $r_h(s_h, a_h)$, and causes the environment to transit to $s_{h+1} \sim \mathbb{P}_h(\cdot \mid s_h, a_h)$. Without loss of generality, we assume there is a terminating state $s_{\text{end}}$ which the environment will *always* transit to at step $H + 1$, and the episode terminates when $s_{\text{end}}$ is reached.

**Policy and value functions**  A (deterministic) policy $\pi$ is a collection of $H$ functions $\{\pi_h : \mathcal{S} \to \mathcal{A}\}_{h=1}^{H}$. We denote $V_h^\pi : \mathcal{S} \to \mathbb{R}$ as the value function at step $h$ for policy $\pi$, so that $V_h^\pi(s)$ gives the expected sum of the remaining rewards received under policy $\pi$, starting from $s_h = s$, till the end of the episode. In symbol,

$$V_h^\pi(s) := \mathbb{E}_\pi \big[ \sum_{h'=h}^{H} r_{h'}(s_{h'}, a_{h'}) \mid s_h = s \big].$$

Similarly, we denote $Q_h^\pi : \mathcal{S} \times \mathcal{A} \to \mathbb{R}$ as the $Q$-value function at step $h$ for policy $\pi$, where

$$Q_h^\pi(s, a) := \mathbb{E}_\pi \big[ \sum_{h'=h}^{H} r_{h'}(s_{h'}, a_{h'}) \mid s_h = s, a_h = a \big].$$

There exists an optimal policy $\pi^\star$, which gives the optimal value function for all states [43], in the sense, $V_h^{\pi^\star}(s) = \sup_\pi V_h^\pi(s)$ for all $h \in [H]$ and $s \in \mathcal{S}$. For notational simplicity, we abbreviate $V^{\pi^\star}$ as $V^\star$. We similarly define the optimal $Q$-value function as $Q^\star$. Recall that $Q^\star$ satisfies the Bellman optimality equation:

$$Q_h^\star(s, a) = (\mathcal{T}_h Q_{h+1}^\star)(s, a) := r_h(s, a) + \mathbb{E}_{s' \sim \mathbb{P}_h(\cdot \mid s, a)} \max_{a' \in \mathcal{A}} Q_{h+1}^\star(s', a'). \tag{1}$$

for all $(s, a, h) \in \mathcal{S} \times \mathcal{A} \times [H]$. We also call $\mathcal{T}_h$ the *Bellman operator* at step $h$.

---

[2]We study deterministic reward for notational simplicity. Our results readily generalize to random rewards.

$\epsilon$**-optimality and regret**   We say a policy $\pi$ is $\epsilon$-optimal if $V_1^\pi(s_1) \geq V_1^\star(s_1) - \epsilon$. Suppose an agent interacts with the environment for $K$ episodes. Denote by $\pi^k$ the policy the agent follows in episode $k \in [K]$. The (accumulative) regret is defined as

$$\text{Reg}(K) := \sum_{k=1}^{K}[V_1^\star(s_1) - V_1^{\pi^k}(s_1)].$$

The objective of reinforcement learning is to find an $\epsilon$-optimal policy within a small number of interactions or to achieve sublinear regret.

## 2.1   Function approximation

In this paper, we consider reinforcement learning with value function approximation. Formally, the learner is given a function class $\mathcal{F} = \mathcal{F}_1 \times \cdots \times \mathcal{F}_H$, where $\mathcal{F}_h \subseteq (\mathcal{S} \times \mathcal{A} \to [0,1])$ offers a set of candidate functions to approximate $Q_h^\star$—the optimal $Q$-value function at step $h$. Since no reward is collected in the $(H+1)^{\text{th}}$ steps, we always set $f_{H+1} = 0$.

Reinforcement learning with function approximation in general is extremely challenging without further assumptions (see, e.g., hardness results in [16, 44]). Below, we present two assumptions about function approximation that are commonly adopted in the literature.

**Assumption 1** (Realizability). *$Q_h^\star \in \mathcal{F}_h$ for all $h \in [H]$.*

Realizability requires the function class is well-specified, i.e., function class $\mathcal{F}$ in fact contains the optimal $Q$-value function $Q^\star$ with no approximation error.

**Assumption 2** (Completeness). *$\mathcal{T}_h \mathcal{F}_{h+1} \subseteq \mathcal{F}_h$ for all $h \in [H]$.*

Note $\mathcal{T}_h \mathcal{F}_{h+1}$ is defined as $\{\mathcal{T}_h f_{h+1} : f_{h+1} \in \mathcal{F}_{h+1}\}$. Completeness requires the function class $\mathcal{F}$ to be closed under the Bellman operator.

When function class $\mathcal{F}$ has finite elements, we can use its cardinality $|\mathcal{F}|$ to measure the "size" of function class $\mathcal{F}$. When addressing function classes with infinite elements, we need a notion similar to cardinality. We use the standard $\epsilon$-covering number.

**Definition 3** ($\epsilon$-covering number). *The $\epsilon$-covering number of a set $\mathcal{V}$ under metric $\rho$, denoted as $\mathcal{N}(\mathcal{V}, \epsilon, \rho)$, is the minimum integer $n$ such that there exists a subset $\mathcal{V}_o \subset \mathcal{V}$ with $|\mathcal{V}_o| = n$, and for any $x \in \mathcal{V}$, there exists $y \in \mathcal{V}_o$ such that $\rho(x, y) \leq \epsilon$.*

We refer readers to standard textbooks [see, e.g., 45] for further properties of covering number. In this paper, we will always apply the covering number on function class $\mathcal{F} = \mathcal{F}_1 \times \cdots \times \mathcal{F}_H$, and use metric $\rho(f, g) = \max_h \|f_h - g_h\|_\infty$. For notational simplicity, we omit the metric dependence and denote the covering number as $\mathcal{N}_\mathcal{F}(\epsilon)$.

## 2.2   Eluder dimension

One class of functions highly related to this paper is the function class of low Eluder dimension [18].

**Definition 4** ($\epsilon$-independence between points). *Let $\mathcal{G}$ be a function class defined on $\mathcal{X}$, and $z, x_1, x_2, \ldots, x_n \in \mathcal{X}$. We say $z$ is $\epsilon$-independent of $\{x_1, x_2, \ldots, x_n\}$ with respect to $\mathcal{G}$ if there exist $g_1, g_2 \in \mathcal{G}$ such that $\sqrt{\sum_{i=1}^n (g_1(x_i) - g_2(x_i))^2} \leq \epsilon$, but $g_1(z) - g_2(z) > \epsilon$.*

Intuitively, $z$ is independent of $\{x_1, x_2, \ldots, x_n\}$ means if that there exist two "certifying" functions $g_1$ and $g_2$, so that their function values are similar at all points $\{x_i\}_{i=1}^n$, but the values are rather different at $z$. This independence relation naturally induces the following complexity measure.

**Definition 5** (Eluder dimension). *Let $\mathcal{G}$ be a function class defined on $\mathcal{X}$. The Eluder dimension $\dim_{\text{E}}(\mathcal{G}, \epsilon)$ is the length of the longest sequence $\{x_1, \ldots, x_n\} \subset \mathcal{X}$ such that there exists $\epsilon' \geq \epsilon$ where $x_i$ is $\epsilon'$-independent of $\{x_1, \ldots, x_{i-1}\}$ for all $i \in [n]$.*

Recall that a vector space has dimension $d$ if and only if $d$ is the length of the longest sequence of elements $\{x_1, \ldots, x_d\}$ such that $x_i$ is linearly independent of $\{x_1, \ldots, x_{i-1}\}$ for all $i \in [n]$. Eluder dimension generalizes the linear independence relation in standard vector space to capture both nonlinear independence and approximate independence, and thus is more general.

# 3 Bellman Eluder Dimension

In this section, we introduce our new complexity measure—Bellman Eluder (BE) dimension. As one of its most important properties, we will show that the family of problems with low BE dimension contains the two existing most general tractable problem classes in RL—problems with low Bellman rank, and problems with low Eluder dimension (see Figure 1).

We start by developing a new distributional version of the original Eluder dimension proposed by Russo and Van Roy [18] (see Section 2.2 for more details).

**Definition 6** ($\epsilon$-independence between distributions). *Let $\mathcal{G}$ be a function class defined on $\mathcal{X}$, and $\nu, \mu_1, \ldots, \mu_n$ be probability measures over $\mathcal{X}$. We say $\nu$ is $\epsilon$-independent of $\{\mu_1, \mu_2, \ldots, \mu_n\}$ with respect to $\mathcal{G}$ if there exists $g \in \mathcal{G}$ such that $\sqrt{\sum_{i=1}^{n}(\mathbb{E}_{\mu_i}[g])^2} \leq \epsilon$, but $|\mathbb{E}_\nu[g]| > \epsilon$.*

**Definition 7** (Distributional Eluder (DE) dimension). *Let $\mathcal{G}$ be a function class defined on $\mathcal{X}$, and $\Pi$ be a family of probability measures over $\mathcal{X}$. The distributional Eluder dimension $\dim_{\mathrm{DE}}(\mathcal{G}, \Pi, \epsilon)$ is the length of the longest sequence $\{\rho_1, \ldots, \rho_n\} \subset \Pi$ such that there exists $\epsilon' \geq \epsilon$ where $\rho_i$ is $\epsilon'$-independent of $\{\rho_1, \ldots, \rho_{i-1}\}$ for all $i \in [n]$.*

Definition 6 and Definition 7 generalize Definition 4 and Definition 5 to their distributional versions, by inspecting the expected values of functions instead of the function values at points, and by restricting the candidate distributions to a certain family $\Pi$. The main advantage of this generalization is exactly in the statistical setting, where estimating the expected values of functions with respect to a certain distribution family can be easier than estimating function values at each point (which is the case for RL in large state spaces).

It is clear that the standard Eluder dimension is a special case of the distributional Eluder dimension, because if we choose $\Pi = \{\delta_x(\cdot) \mid x \in \mathcal{X}\}$ where $\delta_x(\cdot)$ is the dirac measure centered at $x$, then $\dim_{\mathrm{E}}(\mathcal{G}, \epsilon) = \dim_{\mathrm{DE}}(\mathcal{G} - \mathcal{G}, \Pi, \epsilon)$ where $\mathcal{G} - \mathcal{G} = \{g_1 - g_2 : g_1, g_2 \in \mathcal{G}\}$.

Now we are ready to introduce the key notion in this paper—Bellman Eluder dimension.

**Definition 8** (Bellman Eluder (BE) dimension). *Let $(I - \mathcal{T}_h)\mathcal{F} := \{f_h - \mathcal{T}_h f_{h+1} : f \in \mathcal{F}\}$ be the set of Bellman residuals induced by $\mathcal{F}$ at step $h$, and $\Pi = \{\Pi_h\}_{h=1}^{H}$ be a collection of $H$ probability measure families over $\mathcal{S} \times \mathcal{A}$. The $\epsilon$-Bellman Eluder of $\mathcal{F}$ with respect to $\Pi$ is defined as*

$$\dim_{\mathrm{BE}}(\mathcal{F}, \Pi, \epsilon) := \max_{h \in [H]} \dim_{\mathrm{DE}}\big((I - \mathcal{T}_h)\mathcal{F}, \Pi_h, \epsilon\big).$$

**Remark 9** (Q-type v.s. V-type). *Definition 8 is based on the Bellman residuals functions that take a state-action pair as input, thus referred to as Q-type BE dimension. Alternatively, one can define V-type BE dimension using a different set of Bellman residual functions that depend on states only (see Appendix B). We focus on Q-type in the main paper, and present the results for V-type in Appendix B. Both variants are important, and they include different sets of examples (see Appendix B, C).*

In short, Bellman Eluder dimension is simply the distributional Eluder dimension on the function class of Bellman residuals, maximizing over all steps. In addition to function class $\mathcal{F}$ and error $\epsilon$, Bellman Eluder dimension also depends on the choice of distribution family $\Pi$. For the purpose of this paper, we focus on the following two specific choices.

1. $\mathcal{D}_\mathcal{F} := \{\mathcal{D}_{\mathcal{F},h}\}_{h \in [H]}$, where $\mathcal{D}_{\mathcal{F},h}$ denotes the collection of all probability measures over $\mathcal{S} \times \mathcal{A}$ at the $h^{\mathrm{th}}$ step, which can be generated by executing the greedy policy $\pi_f$ induced by any $f \in \mathcal{F}$, i.e., $\pi_{f,h}(\cdot) = \mathrm{argmax}_{a \in \mathcal{A}} f_h(\cdot, a)$ for all $h \in [H]$.

2. $\mathcal{D}_\Delta := \{\mathcal{D}_{\Delta,h}\}_{h \in [H]}$, where $\mathcal{D}_{\Delta,h} = \{\delta_{(s,a)}(\cdot) | s \in \mathcal{S}, a \in \mathcal{A}\}$, i.e., the collections of probability measures that put measure 1 on a single state-action pair.

We say a RL problem has low BE dimension if $\min_{\Pi \in \{\mathcal{D}_\mathcal{F}, \mathcal{D}_\Delta\}} \dim_{\mathrm{BE}}(\mathcal{F}, \Pi, \epsilon)$ is small.

## 3.1 Relations with known tractable classes of RL problems

Known tractable problem classes in RL include but not limited to tabular MDPs, linear MDPs [7], linear quadratic regulators [9], generalized linear MDPs [6], kernel MDPs (Appendix C), reactive POMDPs [16], reactive PSRs [46, 1]. There are two existing generic tractable problem classes that jointly contain all the examples mentioned above: the set of RL problems with low Bellman rank,

and the set of RL problems with low Eluder dimension. However, for these two generic sets, one does not contain the other.

In this section, we will show that our new class of RL problems with low BE dimension in fact contains both low Bellman rank problems and low Eluder dimension problems (see Figure 1). That is, our new problem class covers almost all existing tractable RL problems, and to our best knowledge, is the most generic tractable function class so far.

**Relation with low Bellman rank**    The seminal paper by Jiang et al. [1] proposes the complexity measure—Bellman rank, and shows that a majority of RL examples mentioned above have low Bellman rank. They also propose a hypothesis elimination based algorithm—OLIVE, that learns any low Bellman rank problem within polynomial samples. Formally,

**Definition 10** (Bellman rank). *The Bellman rank is the minimum integer $d$ so that there exists $\phi_h : \mathcal{F} \to \mathbb{R}^d$ and $\psi_h : \mathcal{F} \to \mathbb{R}^d$ for each $h \in [H]$, such that for any $f, f' \in \mathcal{F}$, the average Bellman error.*

$$\mathcal{E}(f, \pi_{f'}, h) := \mathbb{E}_{\pi_{f'}}[(f_h - \mathcal{T}_h f_{h+1})(s_h, a_h)] = \langle \phi_h(f), \psi_h(f') \rangle,$$

*where $\|\phi_h(f)\|_2 \cdot \|\psi_h(f')\|_2 \leq \zeta$, and $\zeta$ is the normalization parameter.*

We remark that similar to Bellman Eluder dimension, Bellman rank also has two variants—Q-type (Definition 10) and V-type (see Appendix B). Recall that we use $\pi_f$ to denote the greedy policy induced by value function $f$. Intuitively, a problem with Bellman rank says its average Bellman error can be decomposed as the inner product of two $d$-dimensional vectors, where one vector depends on the roll-in policy $\pi_{f'}$, while the other vector depends on the value function $f$. At a high level, it claims that the average Bellman error has a linear inner product structure.

**Proposition 11** (low Bellman rank $\subset$ low BE dimension). *If an MDP with function class $\mathcal{F}$ has Bellman rank $d$ with normalization parameter $\zeta$, then*

$$\dim_{\mathrm{BE}}(\mathcal{F}, \mathcal{D}_{\mathcal{F}}, \epsilon) \leq \mathcal{O}(1 + d \log(1 + \zeta/\epsilon)).$$

Proposition 11 claims that problems with low Bellman rank also have low BE dimension, with a small multiplicative factor that is only logarithmic in $\zeta$ and $\epsilon^{-1}$.

**Relation with low Eluder dimension**    Wang et al. [17] study the setting where the function class $\mathcal{F}$ has low Eluder dimension, which includes generalized linear functions. They prove that, when the completeness assumption is satisfied,[3] low Eluder dimension problems can be efficiently learned in polynomial samples.

**Proposition 12** (low Eluder dimension $\subset$ low BE dimension). *Assume $\mathcal{F}$ satisfies completeness (Assumption 2). Then for all $\epsilon > 0$,*

$$\dim_{\mathrm{BE}}\left(\mathcal{F}, \mathcal{D}_{\Delta}, \epsilon\right) \leq \max_{h \in [H]} \dim_{\mathrm{E}}(\mathcal{F}_h, \epsilon).$$

Proposition 12 asserts that problems with low Eluder dimension also have low BE dimension, which is a natural consequence of completeness and the fact that Eluder dimension is a special case of distributional Eluder dimension.

Finally, we show that the set of low BE dimension problems is strictly larger than the union of low Eluder dimension problems and low Bellman rank problems.

**Proposition 13** (low BE dimension $\not\subset$ low Eluder dimension $\cup$ low Bellman rank). *For any $m \in \mathbb{N}^+$, there exists an MDP and a function class $\mathcal{F}$ so that for all $\epsilon \in (0, 1]$, we have $\dim_{\mathrm{BE}}(\mathcal{F}, \mathcal{D}_{\mathcal{F}}, \epsilon) = \dim_{\mathrm{BE}}(\mathcal{F}, \mathcal{D}_{\Delta}, \epsilon) \leq 5$, but $\min\{\min_{h \in [H]} \dim_{\mathrm{E}}(\mathcal{F}_h, \epsilon), \text{Bellman rank}\} \geq m$.*

In particular, the family of low BE dimension includes new examples such as kernel reactive POMDPs (Appendix C), which can not be addressed by the framework of either Bellman rank or Eluder dimension.

---

[3][17] assume for any function $g$ (not necessarily in $\mathcal{F}$), $\mathcal{T}g \in \mathcal{F}$, which is stronger than the completeness assumption presented in this paper (Assumption 2).

**Algorithm 1** GOLF($\mathcal{F}, \mathcal{G}, K, \beta$) — **G**lobal **O**ptimism based on **L**ocal **F**itting

1: **Initialize**: $\mathcal{D}_1, \ldots, \mathcal{D}_H \leftarrow \emptyset$, $\mathcal{B}^0 \leftarrow \mathcal{F}$.
2: **for** episode $k$ from 1 to $K$ **do**
3:     **Choose** policy $\pi^k = \pi_{f^k}$, where $f^k = \operatorname{argmax}_{f \in \mathcal{B}^{k-1}} f(s_1, \pi_f(s_1))$.
4:     **Collect** a trajectory $(s_1, a_1, r_1, \ldots, s_H, a_H, r_H, s_{H+1})$ by following $\pi^k$.
5:     **Augment** $\mathcal{D}_h = \mathcal{D}_h \cup \{(s_h, a_h, r_h, s_{h+1})\}$ for all $h \in [H]$.
6:     **Update**
$$\mathcal{B}^k = \left\{ f \in \mathcal{F} : \ \mathcal{L}_{\mathcal{D}_h}(f_h, f_{h+1}) \leq \inf_{g \in \mathcal{G}_h} \mathcal{L}_{\mathcal{D}_h}(g, f_{h+1}) + \beta \text{ for all } h \in [H] \right\},$$
$$\text{where } \mathcal{L}_{\mathcal{D}_h}(\xi_h, \zeta_{h+1}) = \sum_{(s,a,r,s') \in \mathcal{D}_h} [\xi_h(s,a) - r - \max_{a' \in \mathcal{A}} \zeta_{h+1}(s', a')]^2. \tag{2}$$
7: **Output** $\pi^{\text{out}}$ sampled uniformly at random from $\{\pi^k\}_{k=1}^K$.

# 4   Algorithm GOLF

Section 3 defines a new class of RL problems with low BE dimension, and shows that the new class is rich, containing almost all the existing known tractable RL problems so far. In this section, we propose a new simple optimization-based algorithm—**G**lobal **O**ptimism based on **L**ocal **F**itting (GOLF). We prove that, low BE dimension problems are indeed tractable, i.e., GOLF can find near-optimal policies for these problems within a polynomial number of samples.

At a high level, GOLF can be viewed as an optimistic version of the classic algorithm—Fitted Q-Iteration (FQI) [42]. GOLF generalizes the ELEANOR algorithm [8] from the special linear setting to the general setting with arbitrary function classes.

The pseudocode of GOLF is given in Algorithm 1. GOLF initializes datasets $\{\mathcal{D}_h\}_{h=1}^H$ to be empty sets, and confidence set $\mathcal{B}^0$ to be $\mathcal{F}$. Then, in each episode, GOLF performs two main steps:

- Line 3 (Optimistic planning): compute the most optimistic value function $f^k$ from the confidence set $\mathcal{B}^{k-1}$ constructed in the last episode , and choose $\pi^k$ to be its greedy policy.

- Line 4-6 (Execute the policy and update the confidence set): execute policy $\pi^k$ for one episode, collect data, and update the confidence set using the new data.

At the heart of GOLF is the way we construct the confidence set $\mathcal{B}^k$. For each $h \in [H]$, GOLF maintains a *local* regression constraint using the collected transition data $\mathcal{D}_h$ at this step

$$\mathcal{L}_{\mathcal{D}_h}(f_h, f_{h+1}) \leq \inf_{g \in \mathcal{G}_h} \mathcal{L}_{\mathcal{D}_h}(g, f_{h+1}) + \beta, \tag{3}$$

where $\beta$ is a confidence parameter, and $\mathcal{L}_{\mathcal{D}_h}$ is the squared loss defined in (2), which can be viewed as a proxy to the squared Bellman error at step $h$. We remark that FQI algorithm [42] simply updates $f_h \leftarrow \operatorname{argmin}_{\phi \in \mathcal{F}_h} \mathcal{L}_{\mathcal{D}_h}(\phi, f_{h+1})$. Our constraint (3) can be viewed as a relaxed version of this update, which allows $f_h$ to be not only the minimizer of the loss $\mathcal{L}_{\mathcal{D}_h}(\cdot, f_{h+1})$, but also any function whose loss is only slightly larger than the optimal loss over the auxiliary function class $\mathcal{G}_h$.

We remark that in general, the optimization problem in Line 3 of GOLF can not be solved computationally efficiently.

## 4.1   Theoretical guarantees

In this subsection, we present the theoretical guarantees for GOLF, which hold under Assumption 1 (realizability) and the following generalized completeness assumption introduced in [47, 21]. Let $\mathcal{G} = \mathcal{G}_1 \times \cdots \times \mathcal{G}_H$ be an auxiliary function class provided to the learner where each $\mathcal{G}_h \subseteq (\mathcal{S} \times \mathcal{A} \to [0, 1])$. Generalized completeness requires the auxiliary function class $\mathcal{G}$ to be rich enough so that applying Bellman operator to any function in the primary function class $\mathcal{F}$ will end up in $\mathcal{G}$.

**Assumption 14** (Generalized completeness). $\mathcal{T}_h \mathcal{F}_{h+1} \subseteq \mathcal{G}_h$ *for all* $h \in [H]$.

If we choose $\mathcal{G} = \mathcal{F}$, then Assumption 14 is equivalent to the standard completeness assumption (Assumption 2). Now, we are ready to present the main theorem for GOLF.

**Theorem 15** (Regret of GOLF). *Under Assumption 1, 14, there exists an absolute constant $c$ such that for any $\delta \in (0,1]$, $K \in \mathbb{N}$, if we choose parameter $\beta = c \log[\mathcal{N}_{\mathcal{F} \cup \mathcal{G}}(1/K) \cdot KH/\delta]$ in GOLF, then with probability at least $1 - \delta$, for all $k \in [K]$, we have*

$$\text{Reg}(k) = \sum_{t=1}^{k} \left[ V_1^\star(s_1) - V_1^{\pi^t}(s_1) \right] \leq \mathcal{O}(H\sqrt{dk\beta}),$$

*where $d = \min_{\Pi \in \{\mathcal{D}_\Delta, \mathcal{D}_\mathcal{F}\}} \dim_{\text{BE}} \left( \mathcal{F}, \Pi, 1/\sqrt{K} \right)$ is the BE dimension.*

Theorem 15 asserts that, under the realizability and completeness assumptions, the general class of RL problems with low BE dimension is indeed tractable: there exists an algorithm (GOLF) that can achieve $\sqrt{K}$ regret, whose multiplicative factor depends only polynomially on the horizon of MDP $H$, the BE dimension $d$, and the log covering number of the two function classes. Most importantly, the regret is independent of the number of the states, which is crucial for dealing with practical RL problems with function approximation, where the state spaces are typically exponentially large.

We remark that when function class $\mathcal{F} \cup \mathcal{G}$ has finite number of elements, its covering number is upper bounded by its cardinality $|\mathcal{F} \cup \mathcal{G}|$. For a wide range of function classes in practice, the log $\epsilon'$-covering number has only logarithmic dependence on $\epsilon'$. Informally, we denote the log covering number as $\log \mathcal{N}_{\mathcal{F} \cup \mathcal{G}}$ and omit its $\epsilon'$ dependency for clean presentation. Theorem 15 claims that the regret scales as $\tilde{\mathcal{O}}(H\sqrt{dK \log \mathcal{N}_{\mathcal{F} \cup \mathcal{G}}})$, where $\tilde{\mathcal{O}}(\cdot)$ omits absolute constants and logarithmic terms.[4]

By the standard online-to-batch argument, we also derive the sample complexity of GOLF.

**Corollary 16** (Sample Complexity of GOLF). *Under Assumption 1, 2, there exists an absolute constant $c$ such that for any $\epsilon \in (0,1]$, if we choose $\beta = c \log[\mathcal{N}_{\mathcal{F} \cup \mathcal{G}}(\epsilon^2/(dH^2)) \cdot HK]$ in GOLF, then the output policy $\pi^{out}$ is $\mathcal{O}(\epsilon)$-optimal with probability at least $1/2$, if*

$$K \geq \Omega \left( \frac{H^2 d}{\epsilon^2} \cdot \log \left[ \mathcal{N}_{\mathcal{F} \cup \mathcal{G}} \left( \frac{\epsilon^2}{H^2 d} \right) \cdot \frac{Hd}{\epsilon} \right] \right),$$

*where $d = \min_{\Pi \in \{\mathcal{D}_\Delta, \mathcal{D}_\mathcal{F}\}} \dim_{\text{BE}} \left( \mathcal{F}, \Pi, \epsilon/H \right)$ is the BE dimension.*

Corollary 16 claims that $\tilde{\mathcal{O}}(H^2 d \log(\mathcal{N}_{\mathcal{F} \cup \mathcal{G}})/\epsilon^2)$ samples are enough for GOLF to learn a near-optimal policy of any low BE dimension problem. Our sample complexity scales linear in both the BE dimension $d$, and the log covering number $\log(\mathcal{N}_{\mathcal{F} \cup \mathcal{G}})$.

To showcase the sharpness of our results, we compare them to the previous results when restricted to the corresponding settings. (1) For linear function class with ambient dimension $d_{\text{lin}}$, we have BE dimension $d = \tilde{\mathcal{O}}(d_{\text{lin}})$ and $\log(\mathcal{N}_{\mathcal{F} \cup \mathcal{G}}) = \tilde{\mathcal{O}}(d_{\text{lin}})$. Our regret bound becomes $\tilde{\mathcal{O}}(H d_{\text{lin}} \sqrt{K})$ which matches the best known result [8] up to logarithmic factors; (2) For function class with low Eluder dimension [17], our results hold under weaker completeness assumptions. Our regret scales with $\sqrt{d_\text{E}}$ in terms of dependency on Eluder dimension $d_\text{E}$, which improves the linear $d_\text{E}$ scaling in the regret of [17]; (3) Finally, for low Bellman rank problems, our sample complexity scales linearly with Bellman rank, which improves upon the quadratic dependence in [1]. We remark that all results mentioned above assume (approximate) realizability. All except [1] assume (approximate) completeness.

## 4.2 Key ideas in proving Theorem 15

In this subsection, we present a brief proof sketch for the regret bound of GOLF. We defer all the details to Appendix E. For simplicity, we only discuss the case of choosing $\mathcal{D}_\mathcal{F}$ as the distribution family $\Pi$ in the definition of Bellman Eluder dimension (Definition 8). The proof for using $\mathcal{D}_\Delta$ as the distribution family follows from similar arguments.

Our proof strategy consists of three main steps.

**Step 1: Prove optimism.** We firstly show that, with high probability, the optimal value function $Q^\star$ indeed lies in the confidence set $\mathcal{B}^k$ for all $k \in [K]$ (Lemma 40 in Appendix E.1), which

---

[4]We will not omit $\log \mathcal{N}_{\mathcal{F} \cup \mathcal{G}}$ in $\tilde{\mathcal{O}}(\cdot)$ notation since for many function classes, $\log \mathcal{N}_{\mathcal{F} \cup \mathcal{G}}$ is not small. For instance, for a $\tilde{d}$-dimensional linear function class, $\log \mathcal{N}_{\mathcal{F} \cup \mathcal{G}} = \tilde{\mathcal{O}}(\tilde{d})$.

is a natural consequence of martingale concentration and the properties of the confidence set we designed. Because of $Q^\star \in \mathcal{B}^k$, the optimistic planning step (Line 3) in GOLF guarantees that $V_1^\star(s_1) \leq \max_a f_1^k(s_1, a)$ for every episode $k$. This optimism allows the following upper bound on regret

$$\text{Reg}(K) \leq \sum_{k=1}^{K} \left( \max_a f_1^k(s_1, a) - V_1^{\pi^k}(s_1) \right) = \sum_{h=1}^{H} \sum_{k=1}^{K} \mathbb{E}_{\pi^k} \left[ (f_h^k - \mathcal{T} f_{h+1}^k)(s_h, a_h) \right], \quad (4)$$

where the right equality follows from the standard policy loss decomposition (see, e.g., Lemma 1 in [1]), and $\mathbb{E}_\pi$ denotes the expectation taken over sequence $(s_1, a_1, \ldots, s_H, a_H)$ when executing policy $\pi$.

**Step 2: Utilize the sharpness of our confidence set.**   Recall that our construction of the confidence set in Line 6 of GOLF forces $f^k$ computed in episode $k$ to have a small loss $\mathcal{L}_{\mathcal{D}_h}$, which is a proxy for empirical squared Bellman error under data $\mathcal{D}_h$. Since data $\mathcal{D}_h$ in episode $k$ are collected by executing each $\pi^i$ for one episode for all $i < k$, by standard martingale concentration arguments and the completeness assumption, we can show that with high probability (Lemma 39 in Appendix E.1)

$$\sum_{i=1}^{k-1} \mathbb{E}_{\pi^i} \left[ (f_h^k - \mathcal{T} f_{h+1}^k)(s_h, a_h) \right]^2 \leq \mathcal{O}(\beta), \text{ for all } (k, h) \in [K] \times [H]. \quad (5)$$

**Step 3: Establish relations between (4) and (5).**   So far, we want to upper-bound (4), while we know (5). We note that the RHS of (4) is very similar to the LHS of (5), except that the latter is the squared Bellman error, and the expectation is taken under previous policy $\pi^i$ for $i < k$. To establish the connection between these two, it turns out that we need the Bellman Eluder dimension to be small. Concretely, we have the following lemma.

**Lemma 17.** *Given a function class $\Phi$ defined on $\mathcal{X}$ with $|\phi(x)| \leq 1$ for all $(\phi, x) \in \Phi \times \mathcal{X}$, and a family of probability measures $\Pi$ over $\mathcal{X}$. Suppose sequence $\{\phi_k\}_{k=1}^K \subset \Phi$ and $\{\mu_k\}_{k=1}^K \subset \Pi$ satisfy that for all $k \in [K]$, $\sum_{i=1}^{k-1} (\mathbb{E}_{\mu_i}[\phi_k])^2 \leq \beta$. Then for all $k \in [K]$, $\sum_{i=1}^k |\mathbb{E}_{\mu_i}[\phi_i]| \leq \mathcal{O}(\sqrt{\dim_{\text{DE}}(\Phi, \Pi, 1/k)\beta k})$.*

Lemma 17 is a simplification of Lemma 41 in Appendix E, which is a modification of Lemma 2 in [18]. Intuitively, Lemma 17 can be viewed as an analogue of the pigeon-hole principle for DE dimension. Choose $\Phi$ to be the function class of Bellman residuals, and $\mu_k$ to be the distribution under policy $\pi^k$, we finish the proof.

## 5   Conclusion

In this paper, we propose a new complexity measure—Bellman Eluder (BE) dimension for reinforcement learning with function approximation. Our new complexity measure identifies a new rich class of RL problems that subsumes a majority of existing tractable problem classes in RL. We design a new optimization-based algorithm—GOLF, and provide a new analysis for algorithm OLIVE. Both algorithms show that the new rich class of RL problems we identified in fact can be learned within a polynomial number of samples. We hope our results shed light on the future research in finding the minimal structural assumptions that allow sample-efficient reinforcement learning.

## Acknowledgement

This work was partial supported by National Science Foundation Grant NSF-IIS-2107304.

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
