**Algorithm 2** OLIVE $(\mathcal{F}, \zeta_{\text{act}}, \zeta_{\text{elim}}, n_{\text{act}}, n_{\text{elim}})$

1: **Initialize**: $\mathcal{B}^0 \leftarrow \mathcal{F}, \mathcal{D}_h \leftarrow \emptyset$ for all $h, k$.
2: **for** phase $k = 1, 2, \dots$ **do**
3:     **Choose policy** $\pi^k = \pi_{f^k}$, where $f^k = \arg\max_{f \in \mathcal{B}^{k-1}} f(s_1, \pi_f(s_1))$.
4:     **Execute** $\pi^k$ for $n_{\text{act}}$ episodes and *refresh* $\mathcal{D}_h$ to include the fresh $(s_h, a_h, r_h, s_{h+1})$ tuples.
5:     **Estimate** $\hat{\mathcal{E}}(f^k, \pi^k, h)$ for all $h \in [H]$, where

$$\hat{\mathcal{E}}(g, \pi^k, h) = \frac{1}{|\mathcal{D}_h|} \sum_{(s,a,r,s') \in \mathcal{D}_h} \left( g_h(s, a) - r - \max_{a' \in \mathcal{A}} g_{h+1}(s', a') \right).$$

6:     **if** $\sum_{h=1}^{H} \hat{\mathcal{E}}(f^k, \pi^k, h) \leq H\zeta_{\text{act}}$ **then**
7:         Terminate and output $\pi^k$.
8:     Pick any $t \in [H]$ for which $\hat{\mathcal{E}}(f^k, \pi^k, t) \geq \zeta_{\text{act}}$.
9:     **Execute** $\pi^k$ for $n_{\text{elim}}$ episodes and *refresh* $\mathcal{D}_h$ to include the fresh $(s_h, a_h, r_h, s_{h+1})$ tuples.
10:     **Estimate** $\hat{\mathcal{E}}(f, \pi^k, t)$ for all $f \in \mathcal{F}$.
11:     **Update** $\mathcal{B}^k = \left\{ f \in \mathcal{B}^{k-1} : \left| \hat{\mathcal{E}}(f, \pi^k, t) \right| \leq \zeta_{\text{elim}} \right\}$.

# A   Algorithm OLIVE

In this section, we analyze algorithm OLIVE proposed in [1], which is based on hypothesis elimination. We prove that, despite OLIVE was originally designed for solving low Bellman rank problems, it naturally learns RL problems with low BE dimension as well.

The main advantage of OLIVE comparing to GOLF is that OLIVE does not require the completeness assumption. In return, OLIVE has several disadvantages including worse sample complexity, and no sublinear regret.

The pseudocode of OLIVE is presented in Algorithm 2, where in each phase the algorithm contains the following three main components:

- Line 3 (Optimistic planning): compute the most optimistic value function $f^k$ from the candidate set $\mathcal{B}^{k-1}$, and choose $\pi^k$ to be its greedy policy.

- Line 4-7 (Estimate Bellman error): estimate the Bellman error of $f^k$ under $\pi^k$; output $\pi^k$ if the estimated error is small, and otherwise activate the elimination procedure.

- Line 8-11 (Eliminate functions with large Bellman error): pick a step $t \in [H]$ where the estimated Bellman error exceeds the activation threshold $\zeta_{\text{act}}$; eliminate all functions in the candidate set whose Bellman error at step $t$ exceeds the elimination threshold $\zeta_{\text{elim}}$.

We comment that OLIVE is computationally inefficient in general because implementing the optimistic planning part requires solving an NP-hard problem in the worst case [Theorem 4, 48].

## A.1   Theoretical guarantees

Now, we are ready to present the theoretical guarantee for OLIVE.

**Theorem 18** (OLIVE). *Under Assumption 1, there exists absolute constant $c$ such that if we choose*

$$\zeta_{act} = \frac{2\epsilon}{H}, \; \zeta_{elim} = \frac{\epsilon}{2H\sqrt{d}}, \; n_{act} = \frac{H^2 \iota}{\epsilon^2}, \; and \; n_{elim} = \frac{H^2 d \log(\mathcal{N}_{\mathcal{F}}(\zeta_{elim}/8)) \cdot \iota}{\epsilon^2}$$

*where $d = \dim_{\text{BE}}(\mathcal{F}, \mathcal{D}_{\mathcal{F}}, \epsilon/H)$ and $\iota = c \log(Hd/\delta\epsilon)$, then with probability at least $1 - \delta$, Algorithm 2 will output an $\mathcal{O}(\epsilon)$-optimal policy using at most $\mathcal{O}(H^3 d^2 \log[\mathcal{N}_{\mathcal{F}}(\zeta_{elim}/8)] \cdot \iota/\epsilon^2)$ episodes.*

Theorem 18 claims that OLIVE learns an $\epsilon$-optimal policy of an MDP with BE dimension $d$ within $\tilde{\mathcal{O}}(H^3 d^2 \log(\mathcal{N}_{\mathcal{F}})/\epsilon^2)$ episodes. When specialized to low Bellman rank problems, our sample complexity has the same quadratic dependence on Bellman rank $d$ as in [1].

Comparing to GOLF, the major advantage of OLIVE is that OLIVE does not require completeness assumption (Assumption 2) to work. Nevertheless, OLIVE only learns the RL problems that have

low BE dimension with respect to distribution family $\mathcal{D}_{\mathcal{F}}$, not $\mathcal{D}_\Delta$. The sample complexity of OLIVE is also worse than the sample complexity GOLF (as presented in Corollary 16).

Finally, we comment that interpreting OLIVE through the lens of BE dimension, makes the proof of Theorem 18 surprisingly natural, which follows from the definition of BE dimension along with some standard concentration arguments.

## A.2 Interpret OLIVE with BE dimension

In this subsection, we explain the key idea behind OLIVE through the lens of BE dimension.

To provide a clean high-level view, let us assume all estimates are accurate for now, and the activation threshold $\zeta_{\text{act}}$ and the elimination threshold $\zeta_{\text{elim}}$ satisfy $\zeta_{\text{elim}}\sqrt{d} \leq \zeta_{\text{act}}$, where $d = \dim_{\text{BE}}(\mathcal{F}, \mathcal{D}_{\mathcal{F}}, \zeta_{\text{act}})$. Since $\mathcal{E}(Q^\star, \pi, h) \equiv 0$ for any $(\pi, h)$, $Q^\star$ is always in the candidate set. Therefore, the optimistic planning (Line 3) guarantees $\max_a f_1^k(s_1, a) \geq V_1^\star(s_1)$.

If the Bellman error summation is small (Line 6) i.e., $\sum_{h=1}^H \mathcal{E}(f^k, \pi^k, h) \leq H\zeta_{\text{act}}$, then by simple policy loss decomposition (e.g., Lemma 1 in [1]) and the optimism of $f^k$, $\pi^k$ is $H\zeta_{\text{act}}$-optimal. Otherwise, the elimination procedure is activated at some step $t$ satisfying $\mathcal{E}(f^k, \pi^k, t) \geq \zeta_{\text{act}}$ and all $f$ with $\mathcal{E}(f, \pi^k, t) \geq \zeta_{\text{elim}}$ get eliminated. The *key* observation here is:

> *If the elimination procedure is activated at step $h$ in phase $k_1 < \ldots < k_m$, then the roll-in distribution of $\pi^{k_1}, \ldots, \pi^{k_m}$ at step $h$ is an $\zeta_{\text{act}}$-independent sequence with respect to the class of Bellman residuals $(I - \mathcal{T}_h)\mathcal{F}$ at step $h$. Therefore, we should have $m \leq d$.*

For the sake of contradiction, assume $m \geq d + 1$. Let us prove $\pi^{k_1}, \ldots, \pi^{k_{d+1}}$ is a $\zeta_{\text{act}}$-independent sequence. Firstly, for any $j \in [d+1]$, since $f^{k_j}$ is not eliminated in phase $k_1, \ldots, k_{j-1}$, we have

$$\sqrt{\sum_{i=1}^{j-1} \left(\mathcal{E}(f^{k_j}, \pi^{k_i}, h)\right)^2} \leq \sqrt{d} \times \zeta_{\text{elim}} \leq \zeta_{\text{act}}.$$

Besides, because the elimination procedure is activated at step $h$ in phase $k_j$, we have $\mathcal{E}(f^{k_j}, \pi^{k_j}, h) \geq \zeta_{\text{act}}$. By Definition 6, we obtain that the roll-in distribution of $\pi^{k_j}$ at step $h$ is $\zeta_{\text{act}}$-independent of those of $\pi^{k_1}, \ldots, \pi^{k_{j-1}}$ for $j \in [d+1]$, which contradicts the definition $d = \dim_{\text{BE}}(\mathcal{F}, \mathcal{D}_{\mathcal{F}}, \zeta_{\text{act}})$. As a result, the elimination procedure can happen at most $d$ times for each $h \in [H]$, which means the algorithm should terminate within $dH + 1$ phases and output an $H\zeta_{\text{act}}$-optimal policy.

# B  V-type BE Dimension and Algorithms

The definition of Bellman rank, mentioned in Definition 10 and Proposition 11, is slightly different from the original definition in [1]. We denote the former by **Q-type** and the latter (the original definition) by **V-type**. In this section we introduce V-type BE Dimension as well as V-type variants of GOLF and OLIVE. We show that similar results also hold for the V-type variants.

**Definition 19** (V-type Bellman rank)**.** *The V-type Bellman rank is the minimum integer $d$ so that there exists $\phi_h : \mathcal{F} \to \mathbb{R}^d$ and $\psi_h : \mathcal{F} \to \mathbb{R}^d$ for each $h \in [H]$, such that for any $f, f' \in \mathcal{F}$, the average V-type Bellman error*

$$\mathcal{E}_V(f, \pi_{f'}, h) := \mathbb{E}[(f_h - \mathcal{T}_h f_{h+1})(s_h, a_h) \mid s_h \sim \pi_{f'}, a_h \sim \pi_f] = \langle \phi_h(f), \psi_h(f') \rangle,$$

*where $\|\phi_h(f)\|_2 \cdot \|\psi_h(f')\|_2 \leq \zeta$, and $\zeta$ is the normalization parameter.*

The only difference between these two definitions is how we sample $a_h$. In the Q-type definition we have $a_h \sim \pi_{f'}$ (the roll-in policy), however in the V-type definition we have $a_h \sim \pi_f$ (the greedy policy of the function evaluated in the Bellman error) instead. It is worth mentioning that the Q-type and V-type bellman error coincide whenever $f = f'$; namely, $\mathcal{E}(f, \pi_f, h) = \mathcal{E}_V(f, \pi_f, h)$ for all $f \in \mathcal{F}$.

We can similarly define the V-type variant of BE Dimension. At a high level, **V-type BE dimension** $\dim_{\text{VBE}}(\mathcal{F}, \Pi, \epsilon)$ measures the complexity of finding a function in $\mathcal{F}$ such that its expected Bellman error under any state distribution in $\Pi$ is smaller than $\epsilon$.

**Algorithm 3** V-type GOLF $(\mathcal{F}, K, \beta)$

---

1: **Initialize**: $\mathcal{D}_1, \ldots, \mathcal{D}_H \leftarrow \emptyset, \mathcal{B}^0 \leftarrow \mathcal{F}$.
2: **for** epoch $k$ from 1 to $K$ **do**
3:     **Choose policy** $\pi^k = \pi_{f^k}$, where $f^k = \operatorname{argmax}_{f \in \mathcal{B}^{k-1}} f(s_1, \pi_f(s_1))$.
4:     **for** step $h$ from 1 to $H$ **do**
5:         **Collect** a tuple $(s_h, a_h, r_h, s_{h+1})$ by executing $\pi^k$ at step $1, \ldots, h-1$ and taking action uniformly at random at step $h$.
6:         **Augment** $\mathcal{D}_h = \mathcal{D}_h \cup \{(s_h, a_h, r_h, s_{h+1})\}$ for all $h \in [H]$.
7:     **Update**
$$\mathcal{B}^k = \left\{ f \in \mathcal{F} : \; \mathcal{L}_{\mathcal{D}_h}(f_h, f_{h+1}) \leq \inf_{g \in \mathcal{G}_h} \mathcal{L}_{\mathcal{D}_h}(g, f_{h+1}) + \beta \text{ for all } h \in [H] \right\},$$
$$\text{where } \mathcal{L}_{\mathcal{D}_h}(\xi_h, \zeta_{h+1}) = \sum_{(s,a,r,s') \in \mathcal{D}_h} [\xi_h(s, a) - r - \max_{a' \in \mathcal{A}} \zeta_{h+1}(s', a')]^2.$$
8: **Output** $\pi^{\text{out}}$ sampled uniformly at random from $\{\pi^k\}_{k=1}^K$.

---

**Definition 20** (V-type BE dimension). *Let* $(I - \mathcal{T}_h)V_{\mathcal{F}} \subseteq (\mathcal{S} \to \mathbb{R})$ *be the state-wise Bellman residual class of* $\mathcal{F}$ *at step* $h$ *which is defined as*
$$(I - \mathcal{T}_h)V_{\mathcal{F}} := \left\{ s \mapsto (f_h - \mathcal{T}_h f_{h+1})(s, \pi_{f_h}(s)) : f \in \mathcal{F} \right\}.$$

*Let* $\Pi = \{\Pi_h\}_{h=1}^H$ *be a collection of* $H$ *probability measure families over* $\mathcal{S}$. *The* **V-type $\epsilon$-BE dimension** *of* $\mathcal{F}$ *with respect to* $\Pi$ *is defined as*
$$\dim_{\text{VBE}}(\mathcal{F}, \Pi, \epsilon) := \max_{h \in [H]} \dim_{\text{DE}}\big((I - \mathcal{T}_h)V_{\mathcal{F}}, \Pi_h, \epsilon\big).$$

**Relation with low V-type Bellman rank**    With slight abuse of notation, denote by $\mathcal{D}_{\mathcal{F},h}$ the collection of all probability measures over $\mathcal{S}$ at the $h^{\text{th}}$ step, which can be generated by rolling in with a greedy policy $\pi_f$ with $f \in \mathcal{F}$. Similar to Proposition 11, the following proposition claims that the V-type BE dimension of $\mathcal{F}$ with respect to $\mathcal{D}_{\mathcal{F}} := \{\mathcal{D}_{\mathcal{F},h}\}_{h \in [H]}$ is always upper bounded by its V-type Bellman rank up to some logarithmic factor.

**Proposition 21** (low V-type Bellman rank $\subset$ low V-type BE dimension). *If an MDP with function class* $\mathcal{F}$ *has V-type Bellman rank* $d$ *with normalization parameter* $\zeta$, *then*
$$\dim_{\text{VBE}}(\mathcal{F}, \mathcal{D}_{\mathcal{F}}, \epsilon) \leq \mathcal{O}(1 + d \log(1 + \zeta/\epsilon)).$$

The proof of Proposition 21 is almost the same as that of Proposition 11 in Appendix D.1. We omit it here since the only modification is to replace Q-type Bellman rank with its V-type variant wherever it is used.

## B.1   Algorithm V-type GOLF

In this section we describe the V-type variant of GOLF. The pseudocode is provided in Algorithm 3. Its only difference from the Q-type analogue is in Line 5: for each $h \in [H]$, we roll in with policy $\pi^k$ to sample $s_h$, and then instead of continuing following $\pi^k$ we take random action at step $h$.

Now we present the theoretical guarantee for Algorithm 3. Its proof is almost the same as that of Corollary 16 and can be found in appendix G.2.

**Theorem 22** (V-type GOLF). *Under Assumption 1, 14, there exists an absolute constant* $c$ *such that for any given* $\epsilon > 0$, *if we choose* $\beta = c \log[KHN_{\mathcal{F} \cup \mathcal{G}}(\epsilon^2/(d|\mathcal{A}|H^2))]$, *then with probability at least* $0.99$, $\pi^{\text{out}}$ *is* $\mathcal{O}(\epsilon)$-*optimal, if*
$$K \geq \Omega\left( \frac{H^2 d |\mathcal{A}|}{\epsilon^2} \cdot \log\left[ N_{\mathcal{F} \cup \mathcal{G}}\left( \frac{\epsilon^2}{H^2 d |\mathcal{A}|} \right) \cdot \frac{H d |\mathcal{A}|}{\epsilon} \right] \right),$$
*where* $d = \min_{\Pi \in \{\mathcal{D}_\Delta, \mathcal{D}_{\mathcal{F}}\}} \dim_{\text{VBE}}(\mathcal{F}, \Pi, \epsilon/H)$.

Compared with Theorem 23 (V-type OLIVE), Theorem 22 (V-type GOLF) has the following two advantages.

**Algorithm 4** V-type OLIVE $(\mathcal{F}, \zeta_{\text{act}}, \zeta_{\text{elim}}, n_{\text{act}}, n_{\text{elim}})$

1: **Initialize**: $\mathcal{B}^0 \leftarrow \mathcal{F}, \mathcal{D}_h \leftarrow \emptyset$ for all $h, k$.
2: **for phase** $k = 1, 2, \ldots$ **do**
3:      **Choose policy** $\pi^k = \pi_{f^k}$, where $f^k = \arg\max_{f \in \mathcal{B}^{k-1}} f(s_1, \pi_f(s_1))$.
4:      **Execute** $\pi^k$ for $n_{\text{act}}$ episodes and *refresh* $\mathcal{D}_h$ to include the fresh $(s_h, a_h, r_h, s_{h+1})$ tuples.
5:      **Estimate** $\tilde{\mathcal{E}}_{\text{V}}(f^k, \pi^k, h)$ for all $h \in [H]$, where

$$\tilde{\mathcal{E}}_{\text{V}}(f^k, \pi^k, h) = \frac{1}{|\mathcal{D}_h|} \sum_{(s,a,r,s') \in \mathcal{D}_h} \left( f_h^k(s,a) - r - \max_{a' \in \mathcal{A}} f_{h+1}^k(s', a') \right).$$

6:      **if** $\sum_{h=1}^{H} \tilde{\mathcal{E}}_{\text{V}}(f^k, \pi^k, h) \leq H\zeta_{\text{act}}$ **then**
7:         Terminate and output $\pi^k$.
8:      Pick any $t \in [H]$ for which $\tilde{\mathcal{E}}_{\text{V}}(f^k, \pi^k, t) > \zeta_{\text{act}}$.
9:      **Collect** $n_{\text{elim}}$ episodes by executing $\pi^k$ for step $1, \ldots, t-1$ and picking action uniform at random for step $t$. *Refresh* $\mathcal{D}_h$ to include the fresh $(s_h, a_h, r_h, s_{h+1})$ tuples.
10:      **Estimate** $\hat{\mathcal{E}}_{\text{V}}(f, \pi^k, t)$ for all $f \in \mathcal{F}$, where

$$\hat{\mathcal{E}}_{\text{V}}(f, \pi^k, h) = \frac{1}{|\mathcal{D}_h|} \sum_{(s,a,r,s') \in \mathcal{D}_h} \frac{\mathbf{1}[a = \pi_f(s)]}{1/|\mathcal{A}|} \left( f_h(s,a) - r - \max_{a' \in \mathcal{A}} f_{h+1}(s', a') \right).$$

11:      **Update** $\mathcal{B}^k = \left\{ f \in \mathcal{B}^{k-1} : \left| \hat{\mathcal{E}}_{\text{V}}(f, \pi^k, t) \right| \leq \zeta_{\text{elim}} \right\}$.

---

- The sample complexity in Theorem 22 depends linearly on the V-type BE-dimension while the dependence in Theorem 23 is quadratic.
- Theorem 22 applies to RL problems of finite V-type BE dimension with respect to either $\mathcal{D}_{\mathcal{F}}$ or $\mathcal{D}_{\Delta}$. In comparison, Theorem 23 provides no guarantee for the $\mathcal{D}_{\Delta}$ case.

Finally, we comment that for the low Q-type BE dimension family, we provide both regret and sample complexity guarantees while for the low V-type counterpart, we only derive sample complexity result due to the need of taking actions uniformly at random in Algorithm 4 and Algorithm 3. [38] propose an algorithm that can achieve $\sqrt{T}$-regret for problems of low V-type Bellman rank. It is an interesting open problem to study whether similar techniques can be adapted to the low V-type BE dimension setting so that we can also obtain $\sqrt{T}$-regret.

### B.2   Algorithm V-type OLIVE

In this section, we describe the original OLIVE (i.e., V-type OLIVE) proposed by [1], and its theoretical guarantee in terms of V-type BE dimension.

The pseudocode is provided in Algorithm 4. Its only difference from Algorithm 2 is Line 9-10: note that V-type Bellman rank needs the action at step $t$ to be greedy with respect to the function $f$ instead of being picked by the roll-in policy $\pi^k$, so we choose action $a_t$ uniformly at random and use the importance-weighted estimator to estimate the Bellman error for each $f$.

We have the following similar theoretical guarantee for Algorithm 4. Its proof is almost the same as that of Theorem 18 and can be found in Appendix G.1.

**Theorem 23** (V-type OLIVE). *Assume realizability (Assumption 1) holds and $\mathcal{F}$ is finite. There exists absolute constant $c$ such that if we choose*

$$\zeta_{\text{act}} = \frac{2\epsilon}{H}, \ \zeta_{elim} = \frac{\epsilon}{2H\sqrt{d}}, \ n_{act} = \frac{H^2\iota}{\epsilon^2}, \ and \ n_{elim} = \frac{H^2 d |\mathcal{A}| \log(|\mathcal{F}|) \cdot \iota}{\epsilon^2}$$

*where $d = \dim_{\text{VBE}}(\mathcal{F}, \mathcal{D}_{\mathcal{F}}, \epsilon/H)$ and $\iota = c \log[Hd|\mathcal{A}|/\delta\epsilon]$, then with probability at least $1 - \delta$, Algorithm 4 will output an $\mathcal{O}(\epsilon)$-optimal policy using at most $\mathcal{O}(H^3 d^2 |\mathcal{A}| \log(|\mathcal{F}|) \cdot \iota/\epsilon^2)$ episodes.*

For problems with Bellman rank $d$ and finite function class $\mathcal{F}$, Theorem 23 together with Proposition 21 guarantees $\tilde{\mathcal{O}}(H^3 d^2 |\mathcal{A}| \log(|\mathcal{F}|)/\epsilon^2)$ samples suffice for finding an $\epsilon$-optimal policy, which matches the result in [1]. For function class $\mathcal{F}$ of infinite cardinality but with finite covering number, we can first compute an $\mathcal{O}(\zeta_{\text{elim}})$-cover of $\mathcal{F}$, which we denote as $\mathcal{Z}_{\rho}$, and then run Algorithm 4 on

$\mathcal{Z}_\rho$. By following almost the same arguments in the proof of Theorem 23 (the only difference is to replace $Q^\star$ by its proxy in $\mathcal{Z}_\rho$), we can show Algorithm 4 will output an $\mathcal{O}(\epsilon)$-optimal policy using at most $\tilde{\Omega}(H^3 d^2 |\mathcal{A}| \log(N)/\epsilon^2)$ episodes where $N = \mathcal{N}_\mathcal{F}(\mathcal{O}(\zeta_{\text{elim}}))$.

### B.3 Discussions on Q-type versus V-type

In this paper, we have introduced two complementary definitions of Bellman rank: Q-type Bellman rank and V-type Bellman rank. And we prove they are upper bounds for Q-type and V-type BE dimension, respectively. Here, we want to emphasize that both Q-type and V-type Bellman rank have their own advantages. Specifically, the Q-type version has the following strengths.

1. There are natural RL problems whose Q-type Bellman rank is small, while their V-type Bellman rank is very large, e.g., the linear function approximation setting studied in in [8].

2. All the existing sample complexity results for the V-type cases scale linearly with respect to the number of actions, while those for the Q-type cases are independent of the number of actions. Therefore, for control problems such as Linear Quadratic Regulator (LQR), which has both small Q-type and V-type Bellman rank but infinite number of actions, the notion of Q-type is more suitable.

On the other hand, there are problems that naturally induce low V-type Bellman rank but have large Q-type Bellman rank, e.g., reactive POMDPs.

## C Examples

In this section, we introduce examples with low BE dimension. We will start with linear models and their variants, then introduce kernel MDPs, and finally present kernel reactive POMDPs which have low BE dimension, but possibly large Bellman rank and large Eluder dimension. All the proofs for this section are deferred to Appendix H.

### C.1 Linear models and their variants

In this subsection, we review problems with linear structure in ascending order of generality. We start with the definition of linear MDPs [e.g., 7].

**Definition 24** (Linear MDPs). *We say an MDP is linear of dimension $d$ if for each $h \in [H]$, there exists feature mappings $\phi_h : \mathcal{S} \times \mathcal{A} \to \mathbb{R}^d$, and $d$ unknown signed measures $\psi_h = (\psi_h^{(1)}, \ldots, \psi_h^{(d)})$ over $\mathcal{S}$, and an unknown vector $\theta_h^r \in \mathbb{R}^d$, such that $\mathbb{P}_h(\cdot \mid s, a) = \phi_h(s, a)^\top \psi_h(\cdot)$ and $r_h(s, a) = \phi_h(s, a)^\top \theta_h^r$ for all $(s, a) \in \mathcal{S} \times \mathcal{A}$.*

We remark that existing works [e.g., 7] usually assumxe $\phi$ is *known* to the learner. Next, we review a more general setting—the linear completeness setting [e.g., 8].

**Definition 25** (Linear completeness setting). *We say an MDP is in the linear completeness setting of dimension $d$, if there exists a feature mapping $\phi_h : \mathcal{S} \times \mathcal{A} \to \mathbb{R}^d$, such that for the linear function class $\mathcal{F}_h = \{\phi_h(\cdot)^\top \theta \mid \theta \in \mathbb{R}^d\}$, both Assumption 1 and 2 are satisfied.*

We make three comments here. Firstly, we note that linear MDPs automatically satisfy both linear realizability and linear completeness assumptions, therefore are special cases of the linear completeness setting with the same ambient dimension. Secondly, only assuming linear realizability but without completeness is insufficient for sample-efficient learning (see exponential lower bounds in [44]). Finally, as mentioned in Appendix B.3, though MDPs in the linear completeness setting have low Q-type Bellman rank, their V-type Bellman rank can be arbitrarily large.

Finally, we review the generalized linear completeness setting [6], which generalizes the linear completeness setting by adding nonlinearity.

**Definition 26** (Generalized linear completeness setting). *We say an MDP is in the generalized linear completeness setting of dimension $d$, if there exists a feature mapping $\phi_h : \mathcal{S} \times \mathcal{A} \to \mathbb{R}^d$, and a link function $\sigma$, such that for the generalized linear function class $\mathcal{F}_h = \{\sigma(\phi_h(\cdot)^\top \theta) \mid \theta \in \mathbb{R}^d\}$, both Assumption 1 and 2 are satisfied, and the link function is strictly monotone, i.e., there exist $0 < c_1 < c_2 < \infty$ such that $\sigma'(x) \in [c_1, c_2]$ for all $x$.*

One can directly verify by definition that when we choose link function $\sigma(x) = x$ in the generalized linear completeness setting, it will reduce to the standard linear version. Besides, it is known [18] the generalized linear completeness setting is a special case of low Eluder dimension, thus belonging to the low BE dimension family. Finally, we comment that despite the linear completeness setting belongs to the low Bellman rank family, the generalized version does not because of the possible nonlinearity of the link function.

## C.2 Effective dimension and kernel MDPs

In this subsection, we introduce the notion of effective dimension. With this notion, we prove a useful proposition that any linear kernel function class with low effective dimension also has low Eluder dimension. This proposition directly implies that kernel MDPs are special cases of low Eluder dimension, which are also special cases of low BE dimension.

**Effective dimension** We start with the definition of effective dimension for a set, which is also known as critical information gain in Du et al. [41].

**Definition 27** ($\epsilon$-effective dimension of a set). *The $\epsilon$-effective dimension of a set $\mathcal{X}$ is the minimum integer $d_{\mathrm{eff}}(\mathcal{X}, \epsilon) = n$ such that*

$$\sup_{x_1, \ldots, x_n \in \mathcal{X}} \frac{1}{n} \log \det \left( \mathrm{I} + \frac{1}{\epsilon^2} \sum_{i=1}^{n} x_i x_i^\top \right) \le e^{-1}. \tag{6}$$

Based on this definition, we can also define the effective dimension of a function class.

**Definition 28** ($\epsilon$-effective dimension of a function class). *Given a function class $\mathcal{F}$ defined on $\mathcal{X}$, its $\epsilon$-effective dimension $d_{\mathrm{eff}}(\mathcal{F}, \epsilon) = n$ is the minimum integer $n$ such that there exists a separable Hilbert space $\mathcal{H}$ and a mapping $\phi : \mathcal{X} \to \mathcal{H}$ so that*

- *for every $f \in \mathcal{F}$ there exists $\theta_f \in B_{\mathcal{H}}(1)$ satisfying $f(x) = \langle \theta_f, \phi(x) \rangle_{\mathcal{H}}$ for all $x \in \mathcal{X}$,*
- *$d_{\mathrm{eff}}(\phi(\mathcal{X}), \epsilon) = n$ where $\phi(\mathcal{X}) = \{\phi(x) : x \in \mathcal{X}\}$.*

The following proposition shows that the Eluder dimension of any function class is always upper bounded by its effective dimension.

**Proposition 29** (low effective dimension $\subset$ low Eluder dimension). *For any function class $\mathcal{F}$ and domain $\mathcal{X}$, we have*

$$\dim_{\mathrm{E}}(\mathcal{F}, \epsilon) \le \dim_{\mathrm{eff}}(\mathcal{F}, \epsilon/2).$$

On the other hand, we remark that effective dimension requires the existence of a benign linear structure in certain Hilbert spaces. In constrast, Eluder dimension does not require such conditions. Therefore, the function class of low Eluder dimension is more general than the function class of low effective dimension.

**Kernel MDPs** Now, we are ready to define kernel MDPs and prove it is a subclass of low Eluder dimension.

**Definition 30** (Kernel MDPs). *In a kernel MDP of effective dimension $d(\epsilon)$, for each step $h \in [H]$, there exist feature mappings $\phi_h : \mathcal{S} \times \mathcal{A} \to \mathcal{H}$ and $\psi_h : \mathcal{S} \to \mathcal{H}$ where $\mathcal{H}$ is a separable Hilbert space, so that the transition measure can be represented as the inner product of features, i.e., $\mathbb{P}_h(s' \mid s, a) = \langle \phi_h(s, a), \psi_h(s') \rangle_{\mathcal{H}}$. Besides, the reward function is linear in $\phi$, i.e., $r_h(s, a) = \langle \phi_h(s, a), \theta_h^r \rangle_{\mathcal{H}}$ for some $\theta_h^r \in \mathcal{H}$. Here, $\phi$ is known to the learner while $\psi$ and $\theta^r$ are unknown. Moreover, a kernel MDP satisfies the following regularization conditions: for all $h$*

- *$\|\theta_h^r\|_{\mathcal{H}} \le 1$ and $\|\phi_h(s, a)\|_{\mathcal{H}} \le 1$ for all $s, a$.*
- *$\|\sum_{s \in \mathcal{S}} \mathcal{V}(s) \psi_h(s)\|_{\mathcal{H}} \le 1$ for any function $\mathcal{V} : \mathcal{S} \to [0, 1]$.*
- *$\dim_{\mathrm{eff}}(\mathcal{X}_h, \epsilon) \le d(\epsilon)$ for all $h$ and $\epsilon$, where $\mathcal{X}_h = \{\phi_h(s, a) : (s, a) \in \mathcal{S} \times \mathcal{A}\}$.*

In order to learn kernel MDPs, we need to construct a proper function class $\mathcal{F}$. Formally, for each $h \in [H]$, we choose $\mathcal{F}_h = \{\phi_h(\cdot, \cdot)^\top \theta \mid \theta \in B_{\mathcal{H}}(H + 1 - h)\}$. One can easily verify $\mathcal{F}$ satisfies

both realizability and completeness by following the same arguments as in linear MDPs [7]. In order to apply GOLF or OLIVE, we also need to show it has low BE dimension and bounded log-covering number. Below, we prove in sequence that $\mathcal{F}$ has low Eluder dimension and low log-covering number. Therefore, kernel MDPs fall into our low BE dimension framework.

**Proposition 31** (kernel MDPs $\subset$ low Eluder dimension). *Let $\mathcal{M}$ be a kernel MDP of effective dimension $d(\epsilon)$, then*

$$\dim_{\mathrm{E}}(\mathcal{F}, \epsilon) \leq d(\epsilon/2H).$$

Proposition 31 follows directly from Proposition 29 by rescaling the parameters. Utilizing Proposition 31, we can further prove the log-covering number of $\mathcal{F}$ is also upper bounded by the effective dimension of the kernel MDP up to some logarithmic factor.

**Proposition 32** (bounded covering number). *Let $\mathcal{M}$ be a kernel MDP of effective dimension $d(\epsilon)$, then*

$$\log \mathcal{N}_{\mathcal{F}}(\epsilon) \leq \mathcal{O}\big(Hd(\epsilon) \cdot \log(1 + d(\epsilon)H/\epsilon)\big).$$

### C.3 Effective Bellman rank and kernel reactive POMDPs

To begin with, we introduce the definition of effective Bellman rank and prove that it is always an upper bound for BE dimension. We will see effective Bellman rank serves as a useful tool for controlling the BE dimension of the example discussed in this section—kernel reactive POMDPs.

**Q-type effective Bellman rank**  We start with Q-type $\epsilon$-effective Bellman rank which is simply the $\epsilon$-effective dimension of a special feature set.

**Definition 33** (Q-type $\epsilon$-effective Bellman rank). *The Q-type $\epsilon$-effective Bellman rank is the minimum integer $d$ so that*

- *There exists $\phi_h : \mathcal{F} \to \mathcal{H}$ and $\psi_h : \mathcal{F} \to \mathcal{H}$ for each $h \in [H]$ where $\mathcal{H}$ is a separable Hilbert space, such that for any $f, f' \in \mathcal{F}$, the average Bellman error*

$$\mathcal{E}(f, \pi_{f'}, h) := \mathbb{E}_{\pi_{f'}}[(f_h - \mathcal{T}_h f_{h+1})(s_h, a_h)] = \langle \phi_h(f), \psi_h(f') \rangle_{\mathcal{H}}$$

  *where $\|\phi_h(f)\|_{\mathcal{H}} \leq \zeta$, and $\zeta$ is the normalization parameter.*

- $d = \max_{h \in [H]} d_{\mathrm{eff}}(\mathcal{X}_h(\psi, \mathcal{F}), \epsilon/\zeta)$ *where $\mathcal{X}_h(\psi, \mathcal{F}) = \{\psi_h(f_h) : f_h \in \mathcal{F}_h\}$.*

One can easily verify that when $\mathcal{H}$ is a finite-dimensional Euclidean space, the $\epsilon$-effective Bellman rank is always upper bounded by the original Bellman rank up to a logarithmic factor in $\zeta$ and $\epsilon^{-1}$. Moreover, the effective Bellman rank can be much smaller than the original Bellman rank if the induced feature set $\{\mathcal{X}_h(\psi, \mathcal{F})\}_{h \in [H]}$ approximately lies in a low-dimensional linear subspace. Therefore, effective Bellman rank can be viewed as a strict generalization of the original version.

**Proposition 34** (low Q-type effective Bellman rank $\subset$ low Q-type BE dimension). *Suppose function class $\mathcal{F}$ has Q-type $\epsilon$-effective Bellman rank $d$, then*

$$\dim_{\mathrm{BE}}(\mathcal{F}, \mathcal{D}_{\mathcal{F}}, \epsilon) \leq d.$$

Proposition 34 claims that problems with low Q-type effective Bellman rank also have low Q-type BE dimension.

**V-type effective Bellman rank**  We can similarly define the V-type variant of effective Bellman rank, and prove it is always an upper bound for V-type BE dimension.

**Definition 35** (V-type $\epsilon$-effective Bellman rank). *The V-type $\epsilon$-effective Bellman rank is the minimum integer $d$ so that*

- *There exists $\phi_h : \mathcal{F} \to \mathcal{H}$ and $\psi_h : \mathcal{F} \to \mathcal{H}$ for each $h \in [H]$ where $\mathcal{H}$ is a separable Hilbert space, such that for any $f, f' \in \mathcal{F}$, the average Bellman error*

$$\mathcal{E}_V(f, \pi_{f'}, h) := \mathbb{E}[(f_h - \mathcal{T}_h f_{h+1})(s_h, a_h) \mid s_h \sim \pi_{f'}, a_h \sim \pi_f] = \langle \phi_h(f), \psi_h(f') \rangle_{\mathcal{H}}$$

  *where $\|\phi_h(f)\|_{\mathcal{H}} \leq \zeta$, and $\zeta$ is the normalization parameter.*

- $d = \max_{h \in [H]} d_{\mathrm{eff}}(\mathcal{X}_h(\psi, \mathcal{F}), \epsilon/\zeta)$ *where $\mathcal{X}_h(\psi, \mathcal{F}) = \{\psi_h(f_h) : f_h \in \mathcal{F}_h\}$.*

**Proposition 36** (low V-type effective Bellman rank $\subset$ low V-type BE dimension). *Suppose function class $\mathcal{F}$ has V-type $\epsilon$-effective Bellman rank $d$, then*

$$\dim_{\mathrm{VBE}}(\mathcal{F}, \mathcal{D}_{\mathcal{F}}, \epsilon) \leq d.$$

The proof of Proposition 36 is almost the same as that of Proposition 34. We omit it since the only modification is to replace Q-type effective Bellman rank with its V-type variant wherever it is used.

We want to briefly comment that the majority of examples introduced in [41] have low effective Bellman rank. For example, low occupancy complexity, linear $Q^*/V^*$, linear Bellman complete and $Q^*$ state aggregation have low Q-type effective Bellman rank. And the feature selection problem has low V-type Bellman rank.

**Kernel reactive POMDPs** We start with the definition of POMDPs. A POMDP is defined by a tuple $(\mathcal{S}, \mathcal{A}, \mathcal{O}, \mathbb{T}, \mathbb{O}, r, H)$ where $\mathcal{S}$ denotes the set of hidden states, $\mathcal{A}$ denotes the set of actions, $\mathcal{O}$ denotes the set of observations, $\mathbb{T}$ denotes the transition measure, $\mathbb{O}$ denotes the emission measure, $r = \{r_h\}_{h=1}^{H}$ denotes the collections of reward functions, and $H$ denotes the length of each episode. At the beginning of each episode, the agent always starts from a fixed initial state. At each step $h \in [H]$, after reaching $s_h$, the agent will observe $o_h \sim \mathbb{O}_h(\cdot \mid s_h)$. Then the agent picks action $a_h$, receives $r_h(o_h, a_h)$ and transits to $s_{h+1} \sim \mathbb{T}_h(\cdot \mid s_h, a_h)$. In POMDPs, the agent can never directly observe the states $s_{1:H}$. It can only observe $o_{1:H}$ and $r_{1:H}$. Now we are ready to formally define kernel reactive POMDPs.

**Definition 37** (Kernel reactive POMDPs). *A kernel reactive POMDP is a POMDP that additionally satisfies the following two conditions*

- *For each $h \in [H]$, there exist mappings $\phi_h : \mathcal{S} \times \mathcal{A} \to \mathcal{H}$ and $\psi_h : \mathcal{S} \to \mathcal{H}$ where $\mathcal{H}$ is a separable Hilbert space, such that $\mathbb{T}_h(s' \mid s, a) = \langle \phi_h(s, a), \psi_h(s') \rangle_{\mathcal{H}}$ for all $s', a, s$. Moreover, for any function $\mathcal{V} : \mathcal{S} \to [0, 1]$, $\| \sum_{s' \in \mathcal{S}} \mathcal{V}(s') \psi_h(s') \|_{\mathcal{H}} \leq 1$.*

- *(Reactiveness) The optimal action-value function $Q^*$ only depends on the current observation and action, i.e., for each $h \in [H]$, there exists function $f_h^* : \mathcal{O} \times \mathcal{A} \to [0, 1]$ such that for all $\tau_h = [o_1, a_1, r_1, \ldots, o_h]$ and $a_h$*

$$Q_h^*(\tau_h, a_h) = f_h^*(o_h, a_h).$$

The following proposition shows that when a kernel reactive POMDP has low effective dimension, it also has low V-type BE dimension.

**Proposition 38** (kernel reactive POMDPs $\subset$ low V-type BE dimension). *Any kernel reactive POMDP and function class $\mathcal{F} \subseteq (\mathcal{O} \times \mathcal{A} \to [0, 1])$ satisfy*

$$\dim_{\mathrm{VBE}}(\mathcal{F}, \mathcal{D}_{\mathcal{F}}, \epsilon) \leq \max_{h \in [H]} d_{\mathrm{eff}}(\mathcal{X}_h, \epsilon/2),$$

*where $\mathcal{X}_h = \{\mathbb{E}_{\pi_f}[\phi_h(s_h, a_h)] : f \in \mathcal{F}\}$.*

We comment that when $\mathcal{H}$ *approximately* aligns with a low-dimensional linear subspace, the V-type effective Bellman rank in Proposition 38 will also be low. However, the Eluder dimension of $\mathcal{F}$ can be arbitrarily large because we basically pose no structural assumption on $\mathcal{F}$. Besides, its V/Q-type original Bellman rank can also be arbitrarily large, because $\mathcal{H}$ may be infinite-dimensional and the observation set $\mathcal{O}$ may be exponentially large. If we additionally assume $\mathcal{F}$ satisfies realizability ($f^* \in \mathcal{F}$), then we can apply V-type OLIVE and obtain polynomial sample-complexity guarantee.

## D   Proofs for BE Dimension

In this section, we provide formal proofs for the results stated in Section 3.

### D.1   Proof of Proposition 11

The proof is basically the same as that of Example 3 in [18] with minor modification.

*Proof.* Without loss of generality, assume $\max\{\|\phi_h(f)\|_2, \|\psi_h(f)\|_2\} \le \sqrt{\zeta}$, otherwise we can satisfy this assumption by rescaling the feature mappings. Assume there exists $h \in [H]$ such that $\dim_{\mathrm{DE}}((I - \mathcal{T}_h)\mathcal{F}, \mathcal{D}_{\mathcal{F},h}, \epsilon) \ge m$. Let $\mu_1, \dots, \mu_m \in \mathcal{D}_{\mathcal{F},h}$ be a an $\epsilon$-independent sequence with respect to $(I - \mathcal{T}_h)\mathcal{F}$. By Definition 6, there exists $f^1, \dots, f^m$ such that for all $i \in [m]$, $\sqrt{\sum_{t=1}^{i-1}(\mathbb{E}_{\mu_t}[f_h^i - \mathcal{T}_h f_{h+1}^i])^2} \le \epsilon$ and $|\mathbb{E}_{\mu_i}[f_h^i - \mathcal{T}_h f_{h+1}^i]| > \epsilon$. Since $\mu_1, \dots, \mu_n \in \mathcal{D}_{\mathcal{F},h}$, there exist $g^1, \dots, g^n \in \mathcal{F}$ so that $\mu_i$ is generated by executing $\pi_{g^i}$ for all $i \in [n]$.

By the definition of Bellman rank, this is equivalent to: for all $i \in [m]$, $\sqrt{\sum_{t=1}^{i-1}(\langle \phi_h(g^i), \psi_h(f^t) \rangle)^2} \le \epsilon$ and $|\langle \phi_h(g^i), \psi_h(f^i) \rangle| > \epsilon$.

For notational simplicity, define $\mathbf{x}_i = \phi_h(g^i)$, $\mathbf{z}_i = \psi_h(f^i)$ and $\mathbf{V}_i = \sum_{t=1}^{i-1} \mathbf{z}_t \mathbf{z}_t^\top + \frac{\epsilon^2}{\zeta} \cdot \mathbf{I}$. The previous argument directly implies: for all $i \in [m]$, $\|\mathbf{x}_i\|_{\mathbf{V}_i} \le \sqrt{2}\epsilon$ and $\|\mathbf{x}_i\|_{\mathbf{V}_i} \cdot \|\mathbf{z}_i\|_{\mathbf{V}_i^{-1}} > \epsilon$. Therefore, we have $\|\mathbf{z}_i\|_{\mathbf{V}_i^{-1}} \ge \frac{1}{\sqrt{2}}$.

By the matrix determinant lemma,

$$\det[\mathbf{V}_m] = \det[\mathbf{V}_{m-1}](1 + \|\mathbf{z}_m\|_{\mathbf{V}_m^{-1}}^2) \ge \frac{3}{2}\det[\mathbf{V}_{m-1}] \ge \dots \ge \det[\frac{\epsilon^2}{\zeta} \cdot \mathbf{I}](\frac{3}{2})^{m-1} = (\frac{\epsilon^2}{\zeta})^d (\frac{3}{2})^{m-1}.$$

On the other hand,

$$\det[\mathbf{V}_m] \le (\frac{\mathrm{trace}[\mathbf{V}_m]}{d})^d \le (\frac{\zeta(m-1)}{d} + \frac{\epsilon^2}{\zeta})^d.$$

Therefore, we obtain

$$(\frac{3}{2})^{m-1} \le (\frac{\zeta^2(m-1)}{d\epsilon^2} + 1)^d.$$

Take logarithm on both sides,

$$m \le 4\left[1 + d\log(\frac{\zeta^2(m-1)}{d\epsilon^2} + 1)\right],$$

which, by simple calculation, implies

$$m \le \mathcal{O}\left(1 + d\log(\frac{\zeta^2}{\epsilon^2} + 1)\right). \qquad \square$$

### D.2 Proof of Proposition 12

*Proof.* Assume $\delta_{z_1}, \dots, \delta_{z_m}$ is an $\epsilon$-independent sequence of distributions with respect to $(I - \mathcal{T}_h)\mathcal{F}$, where $\delta_{z_i} \in \mathcal{D}_\Delta$. By Definition 6, there exist functions $f^1, \dots, f^m \in \mathcal{F}$ such that for all $i \in [m]$, we have $|(f_h^i - \mathcal{T}_h f_{h+1}^i)(z_i)| > \epsilon$ and $\sqrt{\sum_{t=1}^{i-1} |(f_h^i - \mathcal{T}_h f_{h+1}^i)(z_t)|^2} \le \epsilon$. Define $g_h^i = \mathcal{T}_h f_{h+1}^i$. Note that $g_h^i \in \mathcal{F}_h$ because $\mathcal{T}_h \mathcal{F}_{h+1} \subset \mathcal{F}_h$. Therefore, we have for all $i \in [m]$, $|(f_h^i - g_h^i)(z_i)| > \epsilon$ and $\sqrt{\sum_{t=1}^{i-1} |(f_h^i - g_h^i)(z_t)|^2} \le \epsilon$ with $f_h^i, g_h^i \in \mathcal{F}_h$. By Definition 4 and 5, this implies $\dim_{\mathrm{E}}(\mathcal{F}_h, \epsilon) \ge m$, which completes the proof. $\square$

### D.3 Proof of Proposition 13

*Proof.* For any $m \in \mathbb{N}^+$, denote by $e_1, \dots, e_m$ the basis vectors in $\mathbb{R}^m$, and consider the following linear bandits ($|\mathcal{S}| = H = 1$) problem.

- The action set $\mathcal{A} = \{a_i = (1; e_i) \in \mathbb{R}^{m+1} : i \in [m]\}$.

- The function set $\mathcal{F}_1 = \{f_{\theta_i}(a) = a^\top \theta_i : \theta_i = (1; e_i), i \in [m]\}$.

- The reward function is always zero, i.e., $r \equiv 0$.

**Eluder dimension** For any $\epsilon \in (0, 1]$, $a_1, \dots, a_{m-1}$ is an $\epsilon$-independent sequence of points because: (a) for any $t \in [m-1]$, $\sum_{i=1}^{t-1}(f_{\theta_t}(a_i) - f_{\theta_{t+1}}(a_i))^2 = 0$; (b) for any $t \in [m-1]$, $f_{\theta_t}(a_t) - f_{\theta_{t+1}}(a_t) = 1 \ge \epsilon$. Therefore, $\min_{h \in [H]} \dim_{\mathrm{E}}(\mathcal{F}_h, \epsilon) = \dim_{\mathrm{E}}(\mathcal{F}_1, \epsilon) \ge m - 1$.

**Bellman rank**   It is direct to see the Bellman residual matrix is $\mathcal{E} := \Theta^\top \Theta \in \mathbb{R}^{m \times m}$ with rank $m$, where $\Theta = [\theta_1, \theta_2, \ldots, \theta_m]$. As a result, the Bellman rank is at least $m$.

**BE dimension**   First, note in this setting $(I - \mathcal{T}_1)\mathcal{F}$ is simply $\mathcal{F}_1$ (because $\mathcal{F}_2 = \{0\}$ and $r \equiv 0$), and $\mathcal{D}_\mathcal{F}$ coincides with $\mathcal{D}_\Delta$, so it suffices to show $\dim_{\mathrm{DE}}(\mathcal{F}_1, \mathcal{D}_\Delta, \epsilon) \leq 5$.

Assume $\dim_{\mathrm{DE}}(\mathcal{F}_1, \mathcal{D}_\Delta, \epsilon) = k$. Then there exist $q_1, \ldots, q_k \in \mathcal{A}$ and $w_1, \ldots, w_k \in \mathcal{A}$ such that for all $t \in [k]$, $\sqrt{\sum_{i=1}^{t-1}(\langle q_t, w_i \rangle)^2} \leq \epsilon$ and $|\langle q_t, w_t \rangle| > \epsilon$. By simple calculation, we have $q_i^\top w_j \in [1, 2]$ for all $i, j \in [k]$. Therefore, if $\epsilon > 2$, then $k = 0$ because $|\langle q_t, w_t \rangle| \leq 2$; if $\epsilon \leq 2$, then $k \leq 5$ because $\sqrt{k-1} \leq \sqrt{\sum_{i=1}^{k-1}(\langle q_k, w_i \rangle)^2} \leq \epsilon$. □

# E   Proofs for GOLF

In this section, we provide formal proofs for the results stated in Section 4.

## E.1   Proof of Theorem 15

We start the proof with the following two lemmas. The first lemma shows that with high probability any function in the confidence set has low Bellman-error over the collected datasets $\mathcal{D}_1, \ldots, \mathcal{D}_H$ as well as the distributions from which $\mathcal{D}_1, \ldots, \mathcal{D}_H$ are sampled.

**Lemma 39.** *Let $\rho > 0$ be an arbitrary fixed number. If we choose $\beta = c\big( \log[KH\mathcal{N}_{\mathcal{F} \cup \mathcal{G}}(\rho)/\delta] + K\rho \big)$ with some large absolute constant $c$ in Algorithm 1, then with probability at least $1 - \delta$, for all $(k, h) \in [K] \times [H]$, we have*

*(a) $\sum_{i=1}^{k-1} \mathbb{E}[\big(f_h^k(s_h, a_h) - (\mathcal{T}f_{h+1}^k)(s_h, a_h)\big)^2 \mid s_h, a_h \sim \pi^i] \leq \mathcal{O}(\beta)$.*

*(b) $\sum_{i=1}^{k-1} \big(f_h^k(s_h^i, a_h^i) - (\mathcal{T}f_{h+1}^k)(s_h^i, a_h^i)\big)^2 \leq \mathcal{O}(\beta)$,*

*where $(s_1^i, a_1^i, \ldots, s_H^i, a_H^i, s_{H+1}^i)$ denotes the trajectory sampled by following $\pi^i$ in the $i^{\mathrm{th}}$ episode.*

The second lemma guarantees that the optimal value function is inside the confidence with high probability. As a result, the selected value function $f^k$ in each iteration shall be an upper bound of $Q^\star$ with high probability.

**Lemma 40.** *Under the same condition of Lemma 39, with probability at least $1 - \delta$, we have $Q^\star \in \mathcal{B}^k$ for all $k \in [K]$.*

The proof of Lemma 39 and 40 relies on standard martingale concentration (e.g. Freedman's inequality) and can be found in Appendix E.3.

**Step 1. Bounding the regret by Bellman error**   By Lemma 40, we can upper bound the cumulative regret by the summation of Bellman error with probability at least $1 - \delta$:

$$\sum_{k=1}^K \left( V_1^\star(s_1) - V_1^{\pi^k}(s_1) \right) \leq \sum_{k=1}^K \left( \max_a f_1^k(s_1, a) - V_1^{\pi^k}(s_1) \right) \overset{(i)}{=} \sum_{k=1}^K \sum_{h=1}^H \mathcal{E}(f^k, \pi^k, h), \quad (7)$$

where $(i)$ follows from standard policy loss decomposition (e.g. Lemma 1 in [1]).

**Step 2. Bounding cumulative Bellman error using DE dimension**   Next, we focus on a fixed step $h$ and bound the cumulative Bellman error $\sum_{k=1}^K \mathcal{E}(f^k, \pi^k, h)$ using Lemma 39. To proceed, we need the following lemma to control the accumulating rate of Bellman error.

**Lemma 41.** *Given a function class $\Phi$ defined on $\mathcal{X}$ with $|\phi(x)| \leq C$ for all $(g, x) \in \Phi \times \mathcal{X}$, and a family of probability measures $\Pi$ over $\mathcal{X}$. Suppose sequence $\{\phi_k\}_{k=1}^K \subset \Phi$ and $\{\mu_k\}_{k=1}^K \subset \Pi$ satisfy that for all $k \in [K]$, $\sum_{t=1}^{k-1}(\mathbb{E}_{\mu_t}[\phi_k])^2 \leq \beta$. Then for all $k \in [K]$ and $\omega > 0$,*

$$\sum_{t=1}^k |\mathbb{E}_{\mu_t}[\phi_t]| \leq \mathcal{O}\left( \sqrt{\dim_{\mathrm{DE}}(\Phi, \Pi, \omega)\beta k} + \min\{k, \dim_{\mathrm{DE}}(\Phi, \Pi, \omega)\}C + k\omega \right).$$

Lemma 41 is a simple modification of Lemma 2 in [18] and its proof can be found in Appendix E.4. We provide two ways to apply Lemma 41, which can produce regret bounds in term of two different complexity measures. If we invoke Lemma 39 (a) and Lemma 41 with

$$
\begin{cases}
\rho = \dfrac{1}{K}, \ \omega = \sqrt{\dfrac{1}{K}}, \ C = 1, \\
\mathcal{X} = \mathcal{S} \times \mathcal{A}, \ \Phi = (I - \mathcal{T}_h)\mathcal{F}, \ \Pi = \mathcal{D}_{\mathcal{F},h}, \\
\phi_k = f_h^k - \mathcal{T}_h f_{h+1}^k \text{ and } \mu_k = \mathbb{P}^{\pi^k}(s_h = \cdot, a_h = \cdot),
\end{cases}
$$

we obtain

$$
\sum_{t=1}^{k} \mathcal{E}(f^t, \pi^t, h) \leq \mathcal{O}\left( \sqrt{k \cdot \dim_{\mathrm{BE}}(\mathcal{F}, \mathcal{D}_{\mathcal{F}}, \sqrt{1/K}) \log[KH\mathcal{N}_{\mathcal{F}\cup\mathcal{G}}(1/K)/\delta]} \right). \tag{8}
$$

We can also invoke Lemma 39 (b) and Lemma 41 with

$$
\begin{cases}
\rho = \dfrac{1}{K}, \ \omega = \sqrt{\dfrac{1}{K}}, \ C = 1, \\
\mathcal{X} = \mathcal{S} \times \mathcal{A}, \ \Phi = (I - \mathcal{T}_h)\mathcal{F}, \text{ and } \Pi = \mathcal{D}_{\Delta,h}, \\
\phi_k = f_h^k - \mathcal{T}_h f_{h+1}^k \text{ and } \mu_k = \mathbf{1}\{\cdot = (s_h^k, a_h^k)\},
\end{cases}
$$

and obtain

$$
\begin{aligned}
\sum_{t=1}^{k} \mathcal{E}(f^t, \pi^t, h) &\leq \sum_{t=1}^{k} (f_h^t - \mathcal{T}f_{h+1}^t)(s_h^t, a_h^t) + \mathcal{O}\left( \sqrt{k \log(k)} \right) \\
&\leq \mathcal{O}\left( \sqrt{k \cdot \dim_{\mathrm{BE}}(\mathcal{F}, \mathcal{D}_{\Delta}, \sqrt{1/K}) \log[KH\mathcal{N}_{\mathcal{F}\cup\mathcal{G}}(1/K)/\delta]} \right),
\end{aligned} \tag{9}
$$

where the first inequality follows from standard martingale concentration.

Plugging either equation (8) or (9) back into equation (7) completes the proof.

## E.2 Proof of Corollary 16

**Step 1. Bounding the regret by Bellman error**  By Lemma 40, we can upper bound the cumulative regret by the summation of Bellman error with probability at least $1 - \delta$:

$$
\sum_{k=1}^{K} \left( V_1^{\star}(s_1) - V_1^{\pi^k}(s_1) \right) \leq \sum_{k=1}^{K} \left( \max_a f_1^k(s_1, a) - V_1^{\pi^k}(s_1) \right) \overset{(i)}{=} \sum_{k=1}^{K} \sum_{h=1}^{H} \mathcal{E}(f^k, \pi^k, h), \tag{10}
$$

where $(i)$ follows from standard policy loss decomposition (e.g. Lemma 1 in [1]).

**Step 2. Bounding cumulative Bellman error using DE dimension**  Next, we focus on a fixed step $h$ and bound the cumulative Bellman error $\sum_{k=1}^{K} \mathcal{E}(f^k, \pi^k, h)$ using Lemma 39.

If we invoke Lemma 39 (a) with

$$
\rho = \frac{\epsilon^2}{H^2 \cdot \dim_{\mathrm{BE}}(\mathcal{F}, \mathcal{D}_{\mathcal{F}}, \epsilon/H)},
$$

and Lemma 41 with

$$
\begin{cases}
\omega = \dfrac{\epsilon}{H}, \ C = 1, \\
\mathcal{X} = \mathcal{S} \times \mathcal{A}, \ \Phi = (I - \mathcal{T}_h)\mathcal{F}, \ \Pi = \mathcal{D}_{\mathcal{F},h}, \\
\phi_k = f_h^k - \mathcal{T}_h f_{h+1}^k \text{ and } \mu_k = \mathbb{P}^{\pi^k}(s_h = \cdot, a_h = \cdot),
\end{cases}
$$

we obtain with probability at least $1 - 10^{-3}$,

$$
\begin{aligned}
\frac{1}{K} \sum_{k=1}^{K} \mathcal{E}(f^k, \pi^k, h) &\leq \mathcal{O}\left( \sqrt{\dim_{\mathrm{BE}}(\mathcal{F}, \mathcal{D}_{\mathcal{F}}, \epsilon/H)[\frac{\log[KH\mathcal{N}_{\mathcal{F}\cup\mathcal{G}}(\rho)]}{K} + \rho]} + \frac{\epsilon}{H} \right) \\
&\leq \mathcal{O}\left( \frac{\epsilon}{H} + \sqrt{\frac{d \log[KH\mathcal{N}_{\mathcal{F}\cup\mathcal{G}}(\rho)]}{K}} \right),
\end{aligned} \tag{11}
$$

where the second inequality follows from the choice of $\rho$ and $d := \dim_{\mathrm{BE}}(\mathcal{F}, \mathcal{D}_\mathcal{F}, \epsilon/H)$. Now we need to choose $K$ such that

$$\sqrt{\frac{d \log[KH\mathcal{N}_{\mathcal{F} \cup \mathcal{G}}(\rho)]}{K}} \leq \frac{\epsilon}{H}. \tag{12}$$

By simple calculation, one can verify it suffices to choose

$$K = \frac{H^2 d \log(Hd\mathcal{N}_{\mathcal{F} \cup \mathcal{G}}(\rho)/\epsilon)}{\epsilon^2}. \tag{13}$$

Plugging equation (11) back into equation (10) completes the proof. We can similarly prove the bound in terms of the BE dimension with respect to $\mathcal{D}_\Delta$.

### E.3 Proofs of concentration lemmas

To begin with, recall the Freedman's inequality that controls the sum of martingale difference by the sum of their predicted variance.

**Lemma 42** (Freedman's inequality [e.g., 49]). *Let $(Z_t)_{t \leq T}$ be a real-valued martingale difference sequence adapted to filtration $\mathfrak{F}_t$, and let $\mathbb{E}_t[\cdot] = \mathbb{E}[\cdot \mid \mathfrak{F}_t]$. If $|Z_t| \leq R$ almost surely, then for any $\eta \in (0, \frac{1}{R})$ it holds that with probability at least $1 - \delta$,*

$$\sum_{t=1}^{T} Z_t \leq \mathcal{O}\left(\eta \sum_{t=1}^{T} \mathbb{E}_{t-1}[Z_t^2] + \frac{\log(\delta^{-1})}{\eta}\right).$$

#### E.3.1 Proof of Lemma 39

*Proof.* We prove inequality $(b)$ first.

Consider a fixed $(k, h, f)$ tuple. Let

$$X_t(h, f) := (f_h(s_h^t, a_h^t) - r_h^t - f_{h+1}(s_{h+1}^t, \pi_f(s_{h+1}^t)))^2 - ((\mathcal{T}f_{h+1})(s_h^t, a_h^t) - r_h^t - f_{h+1}(s_{h+1}^t, \pi_f(s_{h+1}^t)))^2$$

and $\mathfrak{F}_{t,h}$ be the filtration induced by $\{s_1^i, a_1^i, r_1^i, \ldots, s_H^i\}_{i=1}^{t-1} \bigcup \{s_1^t, a_1^t, r_1^t, \ldots, s_h^t, a_h^t\}$. We have

$$\mathbb{E}[X_t(h, f) \mid \mathfrak{F}_{t,h}] = [(f_h - \mathcal{T}f_{h+1})(s_h^t, a_h^t)]^2$$

and

$$\mathrm{Var}[X_t(h, f) \mid \mathfrak{F}_{t,h}] \leq \mathbb{E}[(X_t(h, f))^2 \mid \mathfrak{F}_{t,h}] \leq 36[(f_h - \mathcal{T}f_{h+1})(s_h^t, a_h^t)]^2 = 36\mathbb{E}[X_t(h, f) \mid \mathfrak{F}_{t,h}].$$

By Freedman's inequality, we have, with probability at least $1 - \delta$,

$$\left|\sum_{t=1}^{k} X_t(h, f) - \sum_{t=1}^{k} \mathbb{E}[X_t(h, f) \mid \mathfrak{F}_{t,h}]\right| \leq \mathcal{O}\left(\sqrt{\log(1/\delta) \sum_{t=1}^{k} \mathbb{E}[X_t \mid \mathfrak{F}_{t,h}]} + \log(1/\delta)\right).$$

Let $\mathcal{Z}_\rho$ be a $\rho$-cover of $\mathcal{F}$. Now taking a union bound for all $(k, h, \phi) \in [K] \times [H] \times \mathcal{Z}_\rho$, we obtain that with probability at least $1 - \delta$, for all $(k, h, \phi) \in [K] \times [H] \times \mathcal{Z}_\rho$

$$\left|\sum_{t=1}^{k} X_t(h, \phi) - \sum_{t=1}^{k}[(\phi_h - \mathcal{T}\phi_{h+1})(s_h^t, a_h^t)]^2\right| \leq \mathcal{O}\left(\sqrt{\iota \sum_{t=1}^{k}[(\phi_h - \mathcal{T}\phi_{h+1})(s_h^t, a_h^t)]^2} + \iota\right), \tag{14}$$

where $\iota = \log(HK|\mathcal{Z}_\rho|/\delta)$. From now on, we will do all the analysis conditioning on this event being true.

Consider an arbitrary $(h, k) \in [H] \times [K]$ pair. By the definition of $\mathcal{B}^k$ and Assumption 14

$$
\begin{aligned}
\sum_{t=1}^{k-1} X_t(h, f^k) &= \sum_{t=1}^{k-1} [f_h^k(s_h^t, a_h^t) - r_h^t - f_{h+1}^k(s_{h+1}^t, \pi_{f^k}(s_{h+1}^t))]^2 \\
&\quad - \sum_{t=1}^{k-1} [(\mathcal{T} f_{h+1}^k)(s_h^t, a_h^t) - r_h^t - f_{h+1}^k(s_{h+1}^t, \pi_{f^k}(s_{h+1}^t))]^2 \\
&\leq \sum_{t=1}^{k-1} [f_h^k(s_h^t, a_h^t) - r_h^t - f_{h+1}^k(s_{h+1}^t, \pi_{f^k}(s_{h+1}^t))]^2 \\
&\quad - \inf_{g \in \mathcal{G}} \sum_{t=1}^{k-1} [g_h(s_h^t, a_h^t) - r_h^t - f_{h+1}^k(s_{h+1}^t, \pi_{f^k}(s_{h+1}^t))]^2 \leq \beta.
\end{aligned}
$$

Define $\phi^k = \operatorname{argmin}_{\phi \in \mathcal{Z}_\rho} \max_{h \in [H]} \|f_h^k - \phi_h^k\|_\infty$. By the definition of $\mathcal{Z}_\rho$, we have

$$
\left| \sum_{t=1}^{k-1} X_t(h, f^k) - \sum_{t=1}^{k-1} X_t(h, \phi^k) \right| \leq \mathcal{O}(k\rho).
$$

Therefore,

$$
\sum_{t=1}^{k-1} X_t(h, \phi^k) \leq \mathcal{O}(k\rho) + \beta. \tag{15}
$$

Recall inequality (14) implies

$$
\left| \sum_{t=1}^{k-1} X_t(h, \phi^k) - \sum_{t=1}^{k-1} [(\phi_h^k - \mathcal{T}\phi_{h+1}^k)(s_h^t, a_h^t)]^2 \right| \leq \mathcal{O}\left( \sqrt{\iota \sum_{t=1}^{k-1} [(\phi_h^k - \mathcal{T}\phi_{h+1}^k)(s_h^t, a_h^t)]^2} + \iota \right). \tag{16}
$$

Putting (15) and (16) together, we obtain

$$
\sum_{t=1}^{k-1} [(\phi_h^k - \mathcal{T}\phi_{h+1}^k)(s_h^t, a_h^t)]^2 \leq \mathcal{O}(\iota + k\rho + \beta).
$$

Because $\phi^k$ is an $\rho$-approximation to $f^k$, we conclude

$$
\sum_{t=1}^{k-1} [(f_h^k - \mathcal{T} f_{h+1}^k)(s_h^t, a_h^t)]^2 \leq \mathcal{O}(\iota + k\rho + \beta).
$$

Therefore, we prove inequality $(b)$ in Lemma 39.

To prove inequality $(a)$, we only need to redefine $\mathfrak{F}_{t,h}$ to be the filtration induced by $\{s_1^i, a_1^i, r_1^i, \ldots, s_H^i\}_{i=1}^{t-1}$ and then repeat the arguments above verbatim. $\qquad\square$

### E.3.2 Proof of Lemma 40

*Proof.* Let $\mathcal{V}_\rho$ be a $\rho$-cover of $\mathcal{G}$.

Consider an arbitrary fixed tuple $(k, h, g) \in [K] \times [H] \times \mathcal{G}$. Let

$$
W_t(h, g) := (g_h(s_h^t, a_h^t) - r_h^t - Q_{h+1}^\star(s_{h+1}^t, \pi_{Q^\star}(s_{h+1}^t)))^2 - (Q_h^\star(s_h^t, a_h^t) - r_h^t - Q_{h+1}^\star(s_{h+1}^t, \pi_{Q^\star}(s_{h+1}^t)))^2
$$

and $\mathfrak{F}_{t,h}$ be the filtration induced by $\{s_1^i, a_1^i, r_1^i, \ldots, s_H^i\}_{i=1}^{t-1} \bigcup \{s_1^t, a_1^t, r_1^t, \ldots, s_h^t, a_h^t\}$. We have

$$
\mathbb{E}[W_t(h, g) \mid \mathfrak{F}_{t,h}] = [(g_h - Q_h^\star)(s_h^t, a_h^t)]^2
$$

and

$$
\operatorname{Var}[W_t(h, g) \mid \mathfrak{F}_{t,h}] \leq \mathbb{E}[(W_t(h, g))^2 \mid \mathfrak{F}_{t,h}] \leq 36((g_h - Q_h^\star)(s_h^t, a_h^t))^2 = 36\mathbb{E}[W_t(h, g) \mid \mathfrak{F}_{t,h}].
$$

By Freedman's inequality, with probability at least $1 - \delta$,

$$\left| \sum_{t=1}^{k} W_t(h, g) - \sum_{t=1}^{k} [(g_h - Q_h^\star)(s_h^t, a_h^t)]^2 \right| \leq \mathcal{O}\left( \sqrt{\log(1/\delta) \sum_{t=1}^{k} [(g_h - Q_h^\star)(s_h^t, a_h^t)]^2} + \log(1/\delta) \right).$$

By taking a union bound over $[K] \times [H] \times \mathcal{V}_\rho$ and the non-negativity of $\sum_{t=1}^{k} [(g_h - Q_h^\star)(s_h^t, a_h^t)]^2$, we obtain that with probability at least $1 - \delta$, for all $(k, h, \psi) \in [K] \times [H] \times \mathcal{V}_\rho$

$$- \sum_{t=1}^{k} W_t(h, \psi) \leq \mathcal{O}(\iota),$$

where $\iota = \log(HK|\mathcal{V}_\rho|/\delta)$. This directly implies for all $(k, h, g) \in [K] \times [H] \times \mathcal{G}$

$$\sum_{t=1}^{k-1} [Q_h^\star(s_h^t, a_h^t) - r_h^t - Q_{h+1}^\star(s_{h+1}^t, \pi_{Q^\star}(s_{h+1}^t))]^2$$

$$\leq \sum_{t=1}^{k-1} [g_h(s_h^t, a_h^t) - r_h^t - Q_{h+1}^\star(s_{h+1}^t, \pi_{Q^\star}(s_{h+1}^t))]^2 + \mathcal{O}(\iota + k\rho).$$

Finally, by recalling the definition of $\mathcal{B}^k$, we conclude that with probability at least $1 - \delta$, $Q^\star \in \mathcal{B}^k$ for all $k \in [K]$. $\qquad\square$

### E.4  Proof of Lemma 41

The proof in this subsection basically follows the same arguments as in Appendix C of [18]. We firstly prove the following proposition which bounds the number of times $|\mathbb{E}_{\mu_t}[\phi_t]|$ can exceed a certain threshold.

**Proposition 43.** *Given a function class $\Phi$ defined on $\mathcal{X}$, and a family of probability measures $\Pi$ over $\mathcal{X}$. Suppose sequence $\{\phi_k\}_{k=1}^{K} \subset \Phi$ and $\{\mu_k\}_{k=1}^{K} \subset \Pi$ satisfy that for all $k \in [K]$, $\sum_{t=1}^{k-1}(\mathbb{E}_{\mu_t}[\phi_k])^2 \leq \beta$. Then for all $k \in [K]$,*

$$\sum_{t=1}^{k} \mathbf{1}\{|\mathbb{E}_{\mu_t}[\phi_t]| > \epsilon\} \leq (\frac{\beta}{\epsilon^2} + 1) \dim_{\mathrm{DE}}(\Phi, \Pi, \epsilon).$$

*Proof of Proposition 43.* We first show that if for some $k$ we have $|\mathbb{E}_{\mu_k}[\phi_k]| > \epsilon$, then $\mu_k$ is $\epsilon$-dependent on at most $\beta/\epsilon^2$ disjoint subsequences in $\{\mu_1, \ldots, \mu_{k-1}\}$. By definition of DE dimension, if $|\mathbb{E}_{\mu_k}[\phi_k]| > \epsilon$ and $\mu_k$ is $\epsilon$-dependent on a subsequence $\{\nu_1, \ldots, \nu_\ell\}$ of $\{\mu_1, \ldots, \mu_{k-1}\}$, then we should have $\sum_{t=1}^{\ell}(\mathbb{E}_{\nu_t}[\phi_k])^2 \geq \epsilon^2$. It implies that if $\mu_k$ is $\epsilon$-dependent on $L$ disjoint subsequences in $\{\mu_1, \ldots, \mu_{k-1}\}$, we have

$$\beta \geq \sum_{t=1}^{k-1}(\mathbb{E}_{\mu_t}[\phi_k])^2 \geq L\epsilon^2$$

resulting in $L \leq \beta/\epsilon^2$.

Now we want to show that for any sequence $\{\nu_1, \ldots, \nu_\kappa\} \subseteq \Pi$, there exists $j \in [\kappa]$ such that $\nu_j$ is $\epsilon$-dependent on at least $L = \lceil (\kappa - 1)/\dim_{\mathrm{DE}}(\Phi, \Pi, \epsilon) \rceil$ disjoint subsequences in $\{\nu_1, \ldots, \nu_{j-1}\}$. We argue by the following mental procedure: we start with singleton sequences $B_1 = \{\nu_1\}, \ldots, B_L = \{\nu_L\}$ and $j = L + 1$. For each $j$, if $\nu_j$ is $\epsilon$-dependent on $B_1, \ldots, B_L$ we already achieved our goal so we stop; otherwise, we pick an $i \in [L]$ such that $\nu_j$ is $\epsilon$-independent of $B_i$ and update $B_i = B_i \cup \{\nu_j\}$. Then we increment $j$ by 1 and continue this process. By the definition of DE dimension, the size of each $B_1, \ldots, B_L$ cannot get bigger than $\dim_{\mathrm{DE}}(\Phi, \Pi, \epsilon)$ at any point in this process. Therefore, the process stops before or on $j = L \dim_{\mathrm{DE}}(\Phi, \Pi, \epsilon) + 1 \leq \kappa$.

Fix $k \in [K]$ and let $\{\nu_1, \ldots, \nu_\kappa\}$ be subsequence of $\{\mu_1, \ldots, \mu_k\}$, consisting of elements for which $|\mathbb{E}_{\mu_t}[\phi_t]| > \epsilon$. Using the first claim, we know that each $\nu_j$ is $\epsilon$-dependent on at most $\beta/\epsilon^2$ disjoint

subsequences of $\{\nu_1, \ldots, \nu_{j-1}\}$. Using the second claim, we know there exists $j \in [\kappa]$ such that $\nu_j$ is $\epsilon$-dependent on at least $(\kappa / \dim_{\mathrm{DE}}(\Phi, \Pi, \epsilon)) - 1$ disjoint subsequences of $\{\nu_1, \ldots, \nu_{j-1}\}$. Therefore, we have $\kappa / \dim_{\mathrm{DE}}(\Phi, \Pi, \epsilon) - 1 \leq \beta / \epsilon^2$ which results in

$$\kappa \leq (\frac{\beta}{\epsilon^2} + 1) \dim_{\mathrm{DE}}(\Phi, \Pi, \epsilon)$$

and completes the proof. $\qquad\square$

*Proof of Lemma 41.* Fix $k \in [K]$; let $d = \dim_{\mathrm{DE}}(\Phi, \Pi, \omega)$. Sort the sequence $\{|\mathbb{E}_{\phi_1}[\phi_1]|, \ldots, |\mathbb{E}_{\mu_k}[\phi_k]|\}$ in a decreasing order and denote it by $\{e_1, \ldots, e_k\}$ ($e_1 \geq e_2 \geq \cdots \geq e_k$).

$$\sum_{t=1}^{k} |\mathbb{E}_{\mu_t}[\phi_t]| = \sum_{t=1}^{k} e_t = \sum_{t=1}^{k} e_t \mathbf{1}\{e_t \leq \omega\} + \sum_{t=1}^{k} e_t \mathbf{1}\{e_t > \omega\} \leq k\omega + \sum_{t=1}^{k} e_t \mathbf{1}\{e_t > \omega\}.$$

For $t \in [k]$, we want to prove that if $e_t > \omega$, then we have $e_t \leq \min\{\sqrt{\frac{d\beta}{t-d}}, C\}$. Assume $t \in [k]$ satisfies $e_t > \omega$. Then there exists $\alpha$ such that $e_t > \alpha \geq \omega$. By Proposition 43, we have

$$t \leq \sum_{i=1}^{k} \mathbf{1}\{e_i > \alpha\} \leq (\frac{\beta}{\alpha^2} + 1) \dim_{\mathrm{DE}}(\Phi, \Pi, \alpha) \leq (\frac{\beta}{\alpha^2} + 1) \dim_{\mathrm{DE}}(\Phi, \Pi, \omega),$$

which implies $\alpha \leq \sqrt{\frac{d\beta}{t-d}}$. Besides, recall $e_t \leq C$, so we have $e_t \leq \min\{\sqrt{\frac{d\beta}{t-d}}, C\}$.

Finally, we have

$$\sum_{t=1}^{k} e_t \mathbf{1}\{e_t > \omega\} \leq \min\{d, k\}C + \sum_{t=d+1}^{k} \sqrt{\frac{d\beta}{t-d}} \leq \min\{d, k\}C + \sqrt{d\beta} \int_0^k \frac{1}{\sqrt{t}} dt$$

$$\leq \min\{d, k\}C + 2\sqrt{d\beta k},$$

which completes the proof. $\qquad\square$

# F    Proofs for OLIVE

In this section, we provide the formal proof for the results stated in Appendix A.

## F.1    Full proof of Theorem 18

*Proof of Theorem 18.* By standard concentration arguments (Hoeffding's inequality plus union bound argument), with probability at least $1 - \delta$, the following events hold for the first $dH + 1$ phases (please refer to Appendix F.2 for the proof)

1. If the elimination procedure is activated at the $h^{\mathrm{th}}$ step in the $k^{\mathrm{th}}$ phase, then $\mathcal{E}(f^k, \pi^k, h) > \zeta_{\mathrm{act}}/2$ and all $f \in \mathcal{F}$ satisfying $|\mathcal{E}(f, \pi^k, h)| \geq 2\zeta_{\mathrm{elim}}$ get eliminated.

2. If the elimination procedure is not activated in the $k^{\mathrm{th}}$ phase, then, $\sum_{h=1}^{H} \mathcal{E}(f^k, \pi^k, h) < 2H\zeta_{\mathrm{act}} = 4\epsilon$.

3. $Q^\star$ is not eliminated.

Therefore, if we can show OLIVE terminates within $dH + 1$ phases, then with high probability the output policy is $4\epsilon$-optimal by the optimism of $f^k$ and simple policy loss decomposition (e.g. Lemma 1 in [1]):

$$\left(V_1^\star(s_1) - V_1^{\pi^k}(s_1)\right) \leq \max_a f^k(s_1, a) - V^{\pi^k}(s_1) = \sum_{h=1}^{H} \mathcal{E}(f^k, \pi^k, h) \leq 4\epsilon. \qquad (17)$$

In order to prove that OLIVE terminates within $dH + 1$ phases, it suffices to show that for each $h \in [H]$, we can activate the elimination procedure at the $h^{\mathrm{th}}$ step for at most $d$ times.

For the sake of contradiction, assume that OLIVE does not terminate in $dH + 1$ phases. Within these $dH + 1$ phases, there exists some $h \in [H]$ for which the activation process has been activated for at least $d + 1$ times. Denote by $k_1 < \cdots < k_{d+1} \leq dH + 1$ the indices of the phases where the elimination is activated at the $h^{\text{th}}$ step. By the high-probability events, for all $i < j \leq d + 1$, we have $|\mathcal{E}(f^{k_j}, \pi^{k_i}, h)| < 2\zeta_{\text{elim}}$ and for all $l \leq d + 1$, we have $\mathcal{E}(f^{k_l}, \pi^{k_l}, h) > \zeta_{\text{act}}/2$. This means for all $l \leq d + 1$, we have both $\sqrt{\sum_{i=1}^{l-1} \left(\mathcal{E}(f^{k_l}, \pi^{k_i}, h)\right)^2} < \sqrt{d} \times 2\zeta_{\text{elim}} = \epsilon/H$ and $\mathcal{E}(f^{k_l}, \pi^{k_l}, h) > \zeta_{\text{act}}/2 = \epsilon/H$. Therefore, the roll-in distribution of $\pi^{k_1}, \ldots, \pi^{k_{d+1}}$ at step $h$ is an $\epsilon/H$-independent sequence of length $d + 1$, which contradicts with the definition of BE dimension. So OLIVE should terminate within $dH + 1$ phases.

In sum, with probability at least $1 - \delta$, Algorithm 2 will terminate and output a $4\epsilon$-optimal policy using at most

$$(dH + 1)(n_{\text{act}} + n_{\text{elim}}) \leq \frac{3cH^3 d^2 \log(\mathcal{N}(\mathcal{F}, \zeta_{\text{elim}}/8)) \cdot \iota}{\epsilon^2}$$

episodes. $\qquad \square$

### F.2 Concentration arguments for Theorem 18

Recall in Algorithm 2 we choose

$$\zeta_{\text{act}} = \frac{2\epsilon}{H}, \ \zeta_{\text{elim}} = \frac{\epsilon}{2H\sqrt{d}}, \ n_{\text{act}} = \frac{cH^2 \iota}{\epsilon^2}, \ \text{and} \ n_{\text{elim}} = \frac{cH^2 d \log(\mathcal{N}(\mathcal{F}, \zeta_{\text{elim}}/8)) \cdot \iota}{\epsilon^2},$$

where $d = \max_{h \in [H]} \dim_{\text{BE}}\left(\mathcal{F}, \mathcal{D}_{\mathcal{F}, h}, \epsilon/H\right), \iota = \log[Hd/\delta\epsilon]$ and $c$ is a large absolute constant. Our goal is to prove with probability at least $1 - \delta$, the following events hold for the first $dH + 1$ phases

1. If the elimination procedure is activated at the $h^{\text{th}}$ step in the $k^{\text{th}}$ phase, then $\mathcal{E}(f^k, \pi^k, h) > \zeta_{\text{act}}/2$ and all $f \in \mathcal{F}$ satisfying $|\mathcal{E}(f, \pi^k, h)| \geq 2\zeta_{\text{elim}}$ get eliminated.

2. If the elimination procedure is not activated in the $k^{\text{th}}$ phase, then, $\sum_{h=1}^{H} \mathcal{E}(f^k, \pi^k, h) < 2H\zeta_{\text{act}} = 4\epsilon$.

3. $Q^\star$ is not eliminated.

We begin with the activation procedure.

**Concentration in the activation procedure**  Consider a fixed $(k, h) \in [dH + 1] \times [H]$ pair. By Azuma-Hoefdding's inequality, with probability at least $1 - \frac{\delta}{8H(dH^2+1)}$, we have

$$|\hat{\mathcal{E}}(f^k, \pi^k, h) - \mathcal{E}(f^k, \pi^k, h)| \leq \mathcal{O}\left(\sqrt{\frac{\iota}{n_{\text{act}}}}\right) \leq \frac{\epsilon}{2H} \leq \zeta_{\text{act}}/4,$$

where the second inequality follows from $n_{\text{act}} = C\frac{H^2 \iota}{\epsilon^2}$ with $C$ being chosen large enough.

Take a union bound for all $(k, h) \in [dH + 1] \times [H]$, we have with probability at least $1 - \delta/4$, the following holds for all $(k, h) \in [dH + 1] \times [H]$

$$|\hat{\mathcal{E}}(f^k, \pi^k, h) - \mathcal{E}(f^k, \pi^k, h)| \leq \zeta_{\text{act}}/4.$$

By Algorithm 2, if the elimination procedure is not activated in the $k^{\text{th}}$ phase, we have $\sum_{h=1}^{H} \hat{\mathcal{E}}(f^k, \pi^k, h) \leq H\zeta_{\text{act}}$. Combine it with the concentration argument we just proved,

$$\sum_{h=1}^{H} \mathcal{E}(f^k, \pi^k, h) \leq \sum_{h=1}^{H} \hat{\mathcal{E}}(f^k, \pi^k, h) + \frac{H\zeta_{\text{act}}}{4} < \frac{5H\zeta_{\text{act}}}{4}.$$

On the other hand, if the elimination procedure is activated at the $h^{\text{th}}$ step in the $k^{\text{th}}$ phase, then $\hat{\mathcal{E}}(f^k, \pi^k, h) > \zeta_{\text{act}}$. Again combine it with the concentration argument we just proved,

$$\mathcal{E}(f^k, \pi^k, h) \geq \hat{\mathcal{E}}(f^k, \pi^k, h) - \frac{\zeta_{\text{act}}}{4} > \frac{3\zeta_{\text{act}}}{4}.$$

**Concentration in the elimination procedure**    Now, let us turn to the elimination procedure. First, let $\mathcal{Z}$ be an $\zeta_{\text{elim}}/8$-cover of $\mathcal{F}$ with cardinality $\mathcal{N}(\mathcal{F}, \zeta_{\text{elim}}/8)$. With a little abuse of notation, for every $f \in \mathcal{F}$, define $\hat{f} = \text{argmin}_{g \in \mathcal{Z}} \max_{h \in [H]} \|f_h - g_h\|_\infty$. By applying Azuma-Hoeffding's inequality to all $(k, g) \in [dH + 1] \times \mathcal{Z}$ and taking a union bound, we have with probability at least $1 - \delta/4$, the following holds for all $(k, g) \in [dH + 1] \times \mathcal{Z}$

$$|\hat{\mathcal{E}}(g, \pi^k, h_k) - \mathcal{E}(g, \pi^k, h_k)| \le \zeta_{\text{elim}}/4.$$

Recall that Algorithm 2 eliminates all $f$ satisfying $|\hat{\mathcal{E}}(f, \pi^k, h_k)| > \zeta_{\text{elim}}$ when the elimination procedure is activated at the $h_k^{\text{th}}$ step in the $k^{\text{th}}$ phase. Therefore, if $|\mathcal{E}(f, \pi^k, h_k)| \ge 2\zeta_{\text{elim}}$, $f$ will be eliminated because

$$\begin{aligned}
|\hat{\mathcal{E}}(f, \pi^k, h_k)| &\ge |\hat{\mathcal{E}}(\hat{f}, \pi^k, h_k)| - 2 \times \frac{\zeta_{\text{elim}}}{8} \\
&\ge |\mathcal{E}(\hat{f}, \pi^k, h_k)| - \frac{\zeta_{\text{elim}}}{2} \\
&\ge |\mathcal{E}(f, \pi^k, h_k)| - \frac{\zeta_{\text{elim}}}{2} - 2 \times \frac{\zeta_{\text{elim}}}{8} > \zeta_{\text{elim}}.
\end{aligned}$$

Finally, note that $\mathcal{E}(Q^\star, \pi, h) \equiv 0$ for any $\pi$ and $h$. As a result, it will never be eliminated within the first $dH + 1$ phases because we can similarly prove

$$|\hat{\mathcal{E}}(Q^\star, \pi^k, h_k)| \le |\mathcal{E}(Q^\star, \pi^k, h_k)| + \frac{3\zeta_{\text{elim}}}{4} < \zeta_{\text{elim}}.$$

**Wrapping up**: take a union bound for the activation and elimination procedure, and conclude that the three events, listed at the beginning of this section, hold for the the first $dH + 1$ phases with probability at least $1 - \delta/2$.

## G    Proofs for V-type Variants

In this section, we provide formal proofs for the results stated in Section B.

### G.1    Proof of Theorem 23

The proof is similar to that in Appendix F.

*Proof of Theorem 23.*  By standard concentration arguments (Hoeffding's inequality, Bernstein's inequality, and union bound argument), with probability at least $1 - \delta$, the following events hold for the first $dH + 1$ phases (please refer to Appendix G.1.1 for the proof)

1. If the elimination procedure is activated at the $h^{\text{th}}$ step in the $k^{\text{th}}$ phase, then $\mathcal{E}_{\text{V}}(f^k, \pi^k, h) > \zeta_{\text{act}}/2$ and all $f \in \mathcal{F}$ satisfying $|\mathcal{E}_{\text{V}}(f, \pi^k, h)| \ge 2\zeta_{\text{elim}}$ get eliminated.

2. If the elimination procedure is not activated in the $k^{\text{th}}$ phase, then, $\sum_{h=1}^{H} \mathcal{E}_{\text{V}}(f^k, \pi^k, h) < 2H\zeta_{\text{act}} = 4\epsilon$.

3. $Q^\star$ is not eliminated.

Therefore, if we can show OLIVE terminates within $dH + 1$ phases, then with high probability the output policy is $4\epsilon$-optimal by the optimism of $f^k$ and simple policy loss decomposition (e.g., Lemma 1 in [1]):

$$\left(V_1^\star(s_1) - V_1^{\pi^k}(s_1)\right) \le \max_a f^k(s_1, a) - V^{\pi^k}(s_1) = \sum_{h=1}^{H} \mathcal{E}_{\text{V}}(f^k, \pi^k, h) \le 4\epsilon. \tag{18}$$

In order to prove that OLIVE terminates within $dH + 1$ phases, it suffices to show that for each $h \in [H]$, we can activate the elimination procedure at the $h^{\text{th}}$ step for at most $d$ times.

For the sake of contradiction, assume that OLIVE does not terminate in $dH + 1$ phases. Within these $dH + 1$ phases, there exists some $h \in [H]$ for which the activation process has been activated for at least $d + 1$ times. Denote by $k_1 < \cdots < k_{d+1} \le dH + 1$ the indices of the phases where the elimination is activated at the $h^{\text{th}}$ step. By the high-probability events, for all $i < j \le d + 1$, we have $|\mathcal{E}_{\text{V}}(f^{k_j}, \pi^{k_i}, h)| < 2\zeta_{\text{elim}}$ and for all $l \le d + 1$, we have $\mathcal{E}_{\text{V}}(f^{k_l}, \pi^{k_l}, h) > \zeta_{\text{act}}/2$. This means for all $l \le d + 1$, we have both $\sqrt{\sum_{i=1}^{l-1} \left(\mathcal{E}_{\text{V}}(f^{k_l}, \pi^{k_i}, h)\right)^2} < \sqrt{d} \times 2\zeta_{\text{elim}} = \epsilon/H$ and $\mathcal{E}_{\text{V}}(f^{k_l}, \pi^{k_l}, h) > \zeta_{\text{act}}/2 = \epsilon/H$. Therefore, the roll-in distribution of $\pi^{k_1}, \ldots, \pi^{k_{d+1}}$ at step $h$ is an $\epsilon/H$-independent sequence of length $d + 1$ with respect to $(I - \mathcal{T}_h)V_{\mathcal{F}}$, which contradicts with the definition of BE dimension. So OLIVE should terminate within $dH + 1$ phases.

In sum, with probability at least $1 - \delta$, Algorithm 2 will terminate and output a $4\epsilon$-optimal policy using at most

$$(dH + 1)(n_{\text{act}} + n_{\text{elim}}) \le \frac{3cH^3d^2|\mathcal{A}|\log(|\mathcal{F}|) \cdot \iota}{\epsilon^2}$$

episodes. $\qquad\square$

### G.1.1 Concentration arguments for Theorem 23

Recall in Algorithm 4 we choose

$$\zeta_{\text{act}} = \frac{2\epsilon}{H}, \quad \zeta_{\text{elim}} = \frac{\epsilon}{2H\sqrt{d}}, \quad n_{\text{act}} = \frac{cH^2\iota}{\epsilon^2}, \text{ and } n_{\text{elim}} = \frac{c|\mathcal{A}|H^2 d \log(\mathcal{N}(\mathcal{F}, \zeta_{\text{elim}}/8)) \cdot \iota}{\epsilon^2},$$

where $d = \max_{h \in [H]} \dim_{\text{VBE}}\left(\mathcal{F}, \mathcal{D}_{\mathcal{F}, h}, \epsilon/H\right)$, $\iota = \log[Hd/\delta\epsilon]$ and $c$ is a large absolute constant. Our goal is to prove with probability at least $1 - \delta$, the following events hold for the first $dH + 1$ phases

1. If the elimination procedure is activated at the $h^{\text{th}}$ step in the $k^{\text{th}}$ phase, then $\mathcal{E}_{\text{V}}(f^k, \pi^k, h) > \zeta_{\text{act}}/2$ and all $f \in \mathcal{F}$ satisfying $|\mathcal{E}_{\text{V}}(f, \pi^k, h)| \ge 2\zeta_{\text{elim}}$ get eliminated.

2. If the elimination procedure is not activated in the $k^{\text{th}}$ phase, then, $\sum_{h=1}^{H} \mathcal{E}_{\text{V}}(f^k, \pi^k, h) < 2H\zeta_{\text{act}} = 4\epsilon$.

3. $Q^\star$ is not eliminated.

We begin with the activation procedure.

**Concentration in the activation procedure** Consider a fixed $(k, h) \in [dH + 1] \times [H]$ pair. By Azuma-Hoefdding's inequality, with probability at least $1 - \frac{\delta}{8H(dH+1)}$, we have

$$|\tilde{\mathcal{E}}_{\text{V}}(f^k, \pi^k, h) - \mathcal{E}_{\text{V}}(f^k, \pi^k, h)| \le \mathcal{O}\left(\sqrt{\frac{\iota}{n_{\text{act}}}}\right) \le \frac{\epsilon}{2H} \le \zeta_{\text{act}}/4,$$

where the second inequality follows from $n_{\text{act}} = C\frac{H^2\iota}{\epsilon^2}$ with $C$ being chosen large enough.

Take a union bound for all $(k, h) \in [dH + 1] \times [H]$, we have with probability at least $1 - \delta/4$, the following holds for all $(k, h) \in [dH + 1] \times [H]$

$$|\tilde{\mathcal{E}}_{\text{V}}(f^k, \pi^k, h) - \mathcal{E}_{\text{V}}(f^k, \pi^k, h)| \le \zeta_{\text{act}}/4.$$

By Algorithm 4, if the elimination procedure is not activated in the $k^{\text{th}}$ phase, we have $\sum_{h=1}^{H} \tilde{\mathcal{E}}_{\text{V}}(f^k, \pi^k, h) \le H\zeta_{\text{act}}$. Combine it with the concentration argument we just proved,

$$\sum_{h=1}^{H} \mathcal{E}_{\text{V}}(f^k, \pi^k, h) \le \sum_{h=1}^{H} \tilde{\mathcal{E}}_{\text{V}}(f^k, \pi^k, h) + \frac{H\zeta_{\text{act}}}{4} \le \frac{5H\zeta_{\text{act}}}{4}.$$

On the other hand, if the elimination procedure is activated at the $h^{\text{th}}$ step in the $k^{\text{th}}$ phase, then $\tilde{\mathcal{E}}_{\text{V}}(f^k, \pi^k, h) > \zeta_{\text{act}}$. Again combine it with the concentration argument we just proved,

$$\mathcal{E}_{\text{V}}(f^k, \pi^k, h) \ge \tilde{\mathcal{E}}_{\text{V}}(f^k, \pi^k, h) - \frac{\zeta_{\text{act}}}{4} > \frac{3\zeta_{\text{act}}}{4}.$$

**Concentration in the elimination procedure**  Now, let us turn to the elimination procedure. We start by bounding the the second moment of

$$\frac{\mathbf{1}[\pi_f(s_h) = a_h]}{1/|\mathcal{A}|}\big(f_h(s_h, a_h) - r_h - \max_{a' \in \mathcal{A}} f_{h+1}(s_{h+1}, a')\big)$$

for all $f \in \mathcal{F}$. Let $y(s_h, a_h, r_h, s_{h+1}) = f_h(s_h, a_h) - r_h - \max_{a' \in \mathcal{A}} f_{h+1}(s_{h+1}, a') \in [-2, 1]$, then we have

$$\mathbb{E}[\big(|\mathcal{A}|\mathbf{1}[\pi_f(s_h) = a_h]y(s_h, a_h, r_h, s_{h+1})\big)^2 \mid s_h \sim \pi^k, a_h \sim \text{Uniform}(\mathcal{A})]$$
$$\leq 4|\mathcal{A}|^2 \mathbb{E}[\mathbf{1}[\pi_f(s_h) = a_h] \mid s_h \sim \pi^k, a_h \sim \text{Uniform}(\mathcal{A})] = 4|\mathcal{A}|.$$

For a fixed $(k, f) \in [dH + 1] \times \mathcal{F}$, by applying Azuma-Bernstein's inequality, with probability at least $1 - \frac{\delta}{8(dH+1)|\mathcal{F}|}$ we have

$$|\hat{\mathcal{E}}_{\text{V}}(f, \pi^k, h_k) - \mathcal{E}_{\text{V}}(f, \pi^k, h_k)| \leq \mathcal{O}\left(\sqrt{\frac{|\mathcal{A}|\iota'}{n_{\text{elim}}}} + \frac{|\mathcal{A}|\iota'}{n_{\text{elim}}}\right) \leq \mathcal{O}\left(\sqrt{\frac{|\mathcal{A}|\iota'}{n_{\text{elim}}}}\right) \leq \zeta_{\text{elim}}/2,$$

where $\iota' = \log[8(dH + 1)|\mathcal{F}|/\delta]$, and the third inequality follows from $n_{\text{elim}} = C|\mathcal{A}|\iota/\zeta_{\text{elim}}^2$ with $C$ being chosen large enough.

Taking a union bound over $[dH + 1] \times \mathcal{F}$, we have with probability at least $1 - \delta/4$, the following holds for all $(k, f) \in [dH + 1] \times \mathcal{F}$

$$|\hat{\mathcal{E}}_{\text{V}}(f, \pi^k, h_k) - \mathcal{E}_{\text{V}}(f, \pi^k, h_k)| \leq \zeta_{\text{elim}}/2.$$

Recall that Algorithm 4 eliminates all $f$ satisfying $|\hat{\mathcal{E}}_{\text{V}}(f, \pi^k, h_k)| > \zeta_{\text{elim}}$ when the elimination procedure is activated at the $h_k^{\text{th}}$ step in the $k^{\text{th}}$ phase. Therefore, if $|\mathcal{E}_{\text{V}}(f, \pi^k, h_k)| \geq 2\zeta_{\text{elim}}$, $f$ will be eliminated because

$$|\hat{\mathcal{E}}_{\text{V}}(f, \pi^k, h_k)| \geq |\mathcal{E}_{\text{V}}(f, \pi^k, h_k)| - \frac{\zeta_{\text{elim}}}{2} > \zeta_{\text{elim}}.$$

Finally, note that $\mathcal{E}_{\text{V}}(Q^\star, \pi, h) \equiv 0$ for any $\pi$ and $h$. As a result, it will never be eliminated within the first $dH + 1$ phases because we can similarly prove

$$|\hat{\mathcal{E}}_{\text{V}}(Q^\star, \pi^k, h_k)| \leq |\mathcal{E}_{\text{V}}(Q^\star, \pi^k, h_k)| + \frac{\zeta_{\text{elim}}}{2} < \zeta_{\text{elim}}.$$

**Wrapping up**: take a union bound for the activation and elimination procedure, and conclude that the three events, listed at the beginning of this section, hold for the the first $dH + 1$ phases with probability at least $1 - \delta/2$.

### G.2  Proof of Theorem 22

The proof is basically the same as that of Theorem 15 in Appendix E.

To begin with, we have the following lemma (akin to Lemma 39 and 40) showing that with high probability: $(i)$ any function in the confidence set has low Bellman-error over the collected Datasets $\mathcal{D}_1, \ldots, \mathcal{D}_H$ as well as the distributions from which $\mathcal{D}_1, \ldots, \mathcal{D}_H$ are sampled; $(ii)$ the optimal value function is inside the confidence set. Its proof is almost identical to that of Lemma 39 and 40 which can be found in Appendix E.3.

**Lemma 44** (Akin to Lemma 39 and 40). *Let $\rho > 0$ be an arbitrary fixed number. If we choose $\beta = c\big(\log[KHN_{\mathcal{F} \cup \mathcal{G}}(\rho)/\delta] + K\rho\big)$ with some large absolute constant $c$ in Algorithm 3, then with probability at least $1 - \delta$, for all $(k, h) \in [K] \times [H]$, we have*

*(a)* $\sum_{i=1}^{k-1} \mathbb{E}[\big(f_h^k(s_h, a_h) - (\mathcal{T}f_{h+1}^k)(s_h, a_h)\big)^2 \mid s_h \sim \pi^i, a_h \sim \text{Uniform}(\mathcal{A})] \leq \mathcal{O}(\beta)$,

*(b)* $\frac{1}{|\mathcal{A}|} \sum_{i=1}^{k-1} \sum_{a \in \mathcal{A}} \big(f_h^k(s_h^i, a) - (\mathcal{T}f_{h+1}^k)(s_h^i, a)\big)^2 \leq \mathcal{O}(\beta)$,

*(c)* $Q^\star \in \mathcal{B}^k$,

*where $s_h^i$ denotes the state at step $h$ collected according to Line 5 in Algorithm 3 following $\pi^i$.*

*Proof of Lemma 44.* To prove inequality $(a)$, we only need to redefine the filtration $\mathfrak{F}_{t,h}$ in Appendix E.3.1 to be the filtration induced by $\{s_1^i, a_1^i, r_1^i, \ldots, s_H^i\}_{i=1}^{t-1}$ and repeat the arguments there verbatim.

To prove inequality $(b)$, we only need to redefine the filtration $\mathfrak{F}_{t,h}$ in Appendix E.3.1 to be the filtration induced by $\{s_1^i, a_1^i, r_1^i, \ldots, s_H^i\}_{i=1}^{t-1} \bigcup \{s_1^t, a_1^t, r_1^t, \ldots, s_h^t\}$ and repeat the arguments there verbatim.

The proof of $(c)$ is the same as that of Lemma 40 in Appendix E.3.2. $\qquad\square$

**Step 1. Bounding the regret by Bellman error** By Lemma 44 $(c)$, we can upper bound the cumulative regret by the summation of Bellman error with probability at least $1 - \delta$:

$$\sum_{k=1}^{K} \left( V_1^\star(s_1) - V_1^{\pi^k}(s_1) \right) \leq \sum_{k=1}^{K} \left( \max_a f_1^k(s_1, a) - V_1^{\pi^k}(s_1) \right) \stackrel{(i)}{=} \sum_{k=1}^{K} \sum_{h=1}^{H} \mathcal{E}_{\mathsf{V}}(f^k, \pi^k, h), \quad (19)$$

where $(i)$ follows from standard policy loss decomposition (e.g. Lemma 1 in [1]).

**Step 2. Bounding cumulative Bellman error using DE dimension** Next, we focus on a fixed step $h$ and bound the cumulative Bellman error $\sum_{k=1}^{K} \mathcal{E}_{\mathsf{V}}(f^k, \pi^k, h)$ using Lemma 44.

Invoking Lemma 44 (a) with

$$\rho = \frac{\epsilon^2}{H^2 \cdot \dim_{\mathrm{VBE}}(\mathcal{F}, \mathcal{D}_\mathcal{F}, \epsilon/H) \cdot |\mathcal{A}|}$$

implies that with probability at least $1 - \delta$, for all $(k, h) \in [K] \times [H]$, we have

$$\sum_{i=1}^{k-1} \mathbb{E}\left[ \left( f_h^k(s_h, \pi_{f_h^k}(s_h)) - (\mathcal{T}f_{h+1}^k)(s_h, \pi_{f_h^k}(s_h)) \right)^2 \mid s_h \sim \pi^i \right] \leq \mathcal{O}(|\mathcal{A}|\beta).$$

Further invoking Lemma 41 with

$$\begin{cases} \omega = \dfrac{\epsilon}{H}, \ C = 1, \\ \mathcal{X} = \mathcal{S}, \ \Phi = (I - \mathcal{T}_h)V_\mathcal{F}, \ \Pi = \mathcal{D}_{\mathcal{F},h}, \\ \phi_k(s) := (f_h^k - \mathcal{T}_h f_{h+1}^k)(s, \pi_{f_h^k}(s)) \text{ and } \mu_k = \mathbb{P}^{\pi^k}(s_h = \cdot), \end{cases}$$

we obtain

$$\frac{1}{K} \sum_{t=1}^{K} \mathcal{E}_{\mathsf{V}}(f^t, \pi^t, h) \leq \mathcal{O}\left( \sqrt{\frac{\dim_{\mathrm{VBE}}(\mathcal{F}, \mathcal{D}_\mathcal{F}, \epsilon/H)|\mathcal{A}| \log[KH\mathcal{N}_{\mathcal{F}\cup\mathcal{G}}(\rho)/\delta]}{K}} + \frac{\epsilon}{H} \right).$$

Plugging in the choice of $K$ completes the proof.

Similarly, for $\mathcal{D}_\Delta$, we can invoke Lemma 44 (b) witht

$$\rho = \frac{\epsilon^2}{H^2 \cdot \dim_{\mathrm{VBE}}(\mathcal{F}, \mathcal{D}_\Delta, \epsilon/H) \cdot |\mathcal{A}|},$$

and Lemma 41 with

$$\begin{cases} \omega = \dfrac{\epsilon}{H}, \ C = 1, \\ \mathcal{X} = \mathcal{S}, \ \Phi = (I - \mathcal{T}_h)V_\mathcal{F}, \ \Pi = \mathcal{D}_{\Delta,h}, \\ \phi_k(s) := (f_h^k - \mathcal{T}_h f_{h+1}^k)(s, \pi_{f_h^k}(s)) \text{ and } \mu_k = \mathbf{1}\{\cdot = s_h^k\}, \end{cases}$$

and obtain

$$\frac{1}{K} \sum_{t=1}^{K} \mathcal{E}_{\mathsf{V}}(f^t, \pi^t, h) \leq \frac{1}{K} \sum_{t=1}^{K} (f_h^t - \mathcal{T}f_{h+1}^t)(s_h^t, \pi_{f_h^t}(s_h^t)) + \mathcal{O}\left( \sqrt{\frac{\log K}{K}} \right)$$

$$\leq \mathcal{O}\left( \sqrt{\frac{\dim_{\mathrm{VBE}}(\mathcal{F}, \mathcal{D}_\Delta, \epsilon/H)|\mathcal{A}| \log[KH\mathcal{N}_{\mathcal{F}\cup\mathcal{G}}(\rho)/\delta]}{K}} + \frac{\epsilon}{H} + \sqrt{\frac{\log K}{K}} \right),$$

where the first inequality follows from standard martingale concentration.

Plugging in the choice of $K$ completes the proof.

# H    Proofs for Examples

## H.1    Proof of Proposition 29

*Proof.* Suppose $\mathcal{F}$ has finite $\epsilon$-effective dimension and denote the corresponding mapping by $\phi$. Then we can rewrite $\mathcal{F}$ in the form of $\mathcal{F} = \{f_\theta(\cdot) = \langle \phi(\cdot), \theta \rangle_{\mathcal{H}} \mid \theta \in \Theta\}$, where $\Theta \subset B_{\mathcal{H}}(1)$.

Suppose there exists an $\epsilon'$-independent sequence $x_1', \ldots, x_n' \in \mathcal{X}$ with respect to $\mathcal{F}$ where $\epsilon' \geq \epsilon$. By the definition of independent sequence, this is equivalent to the existence of $\theta_1, \ldots, \theta_n \in (\Theta - \Theta)$ and $x_1, \ldots, x_n \in \phi(\mathcal{X})$ such that

$$\begin{cases} \sum_{i=1}^{t-1} (x_i^\top \theta_t)^2 \leq \epsilon'^2, & t \in [n] \\ |x_t^\top \theta_t| \geq \epsilon', & t \in [n]. \end{cases} \tag{20}$$

Define $\Sigma_t = \sum_{i=1}^{t-1} x_i x_i^\top + \frac{\epsilon'^2}{4} \cdot I$. We have

$$\|\theta_t\|_{\Sigma_t} \leq \sqrt{2}\epsilon' \implies \epsilon' \leq |x_t^\top \theta_t| \leq \|\theta_t\|_{\Sigma_t} \cdot \|x_t\|_{\Sigma_t^{-1}} \leq \sqrt{2}\epsilon' \|x_t\|_{\Sigma_t^{-1}}, \quad t \in [n]. \tag{21}$$

As a result, we should have $\|x_t\|_{\Sigma_t^{-1}}^2 \geq 1/2$ for all $t \in [n]$. Now we can apply the standard log-determinant argument,

$$\sum_{t=1}^n \log(1 + \|x_t\|_{\Sigma_t^{-1}}^2) = \log\left(\frac{\det(\Sigma_{n+1})}{\det(\Sigma_1)}\right) = \log \det\left(I + \frac{4}{\epsilon'^2} \sum_{i=1}^n x_i x_i^\top\right),$$

which implies

$$0.5 \leq \min_{t \in [n]} \|x_t\|_{\Sigma_t^{-1}}^2 \leq \exp\left(\frac{1}{n} \log \det\left(I + \frac{4}{\epsilon'^2} \sum_{i=1}^n x_i x_i^\top\right)\right) - 1. \tag{22}$$

Choose $n = d_{\text{eff}}(\mathcal{F}, \epsilon/2)$ that is the minimum positive integer satisfying

$$\sup_{x_1, \ldots, x_n \in \phi(\mathcal{X})} \frac{1}{n} \log \det\left(I + \frac{4}{\epsilon^2} \sum_{i=1}^n x_i x_i^\top\right) \leq e^{-1}. \tag{23}$$

This leads to a contradiction because $\epsilon' \geq \epsilon$ and $0.5 > e^{e^{-1}} - 1$. So we must have

$$\dim_{\text{E}}(\mathcal{F}, \epsilon) \leq d_{\text{eff}}(\mathcal{F}, \epsilon/2).$$

$\square$

## H.2    Proof of Proposition 32

*Proof.* Consider fixed $\epsilon \in \mathbb{R}^+$ and $h \in [H]$, and denote $n = \dim_{\text{E}}(\mathcal{F}, \epsilon)$. Then by the definition of Eluder dimension, there must exist $x_1, \ldots, x_n \in \mathcal{X}_h$ where $\mathcal{X}_h = \{\phi_h(s, a) : (s, a) \in \mathcal{S} \times \mathcal{A}\}$ so that for any $\theta, \theta' \in B_{\mathcal{H}}(H + 1 - h)$, if $\sum_{i=1}^n (\langle x_i, \theta - \theta' \rangle_{\mathcal{H}})^2 \leq \epsilon^2$, then $|\langle z, \theta - \theta' \rangle_{\mathcal{H}}| \leq \epsilon$ for any $z \in \mathcal{X}_h$. In other words, $x_1, \ldots, x_n$ is one of the longest independent subsequences. Therefore, in order to cover $\mathcal{F}_h$, we only need cover the projection of $B_{\mathcal{H}}(H + 1 - h)$ onto the linear subspace spanned by $x_1, \ldots, x_n$, which is at most $n$ dimensional.

By standard $\epsilon$-net argument, there exists $\mathcal{C} \subset B_{\mathcal{H}}(H + 1 - h)$ such that: (a) $\log|\mathcal{C}| \leq \mathcal{O}(n \cdot \log(1 + nH/\epsilon))$, (b) for any $\theta \in B_{\mathcal{H}}(H + 1 - h)$, there exists $\hat{\theta} \in \mathcal{C}$ satisfying $\sum_{i=1}^n (\langle x_i, \theta - \hat{\theta} \rangle_{\mathcal{H}})^2 \leq \epsilon^2$. By the property of $x_1, \ldots, x_n$, $\{\phi_h(\cdot, \cdot)^\top \hat{\theta} \mid \hat{\theta} \in \mathcal{C}\}$ is an $\epsilon$-cover of $\mathcal{F}_h$. Since $\mathcal{F} = \mathcal{F}_1 \times \cdots \times \mathcal{F}_H$, we obtain $\log \mathcal{N}_{\mathcal{F}}(\epsilon) \leq \mathcal{O}(Hn \cdot \log(1 + nH/\epsilon))$. Finally, by Proposition 31, $n \leq d(\epsilon)$, which concludes the proof.

$\square$

### H.3 Proof of Proposition 34

*Proof.* Assume there exists $h \in [H]$ such that $\dim_{\mathrm{DE}}((I - \mathcal{T}_h)\mathcal{F}, \mathcal{D}_{\mathcal{F},h}, \epsilon) \geq m$. Let $\mu_1, \ldots, \mu_n \in \mathcal{D}_{\mathcal{F},h}$ be a an $\epsilon$-independent sequence with respect to $(I - \mathcal{T}_h)\mathcal{F}$. By Definition 6, there exist $f^1, \ldots, f^n$ such that for all $t \in [n]$, $\sqrt{\sum_{i=1}^{t-1}(\mathbb{E}_{\mu_i}[f_h^t - \mathcal{T}_h f_{h+1}^t])^2} \leq \epsilon$ and $|\mathbb{E}_{\mu_t}[f_h^t - \mathcal{T}_h f_{h+1}^t]| > \epsilon$. Since $\mu_1, \ldots, \mu_n \in \mathcal{D}_{\mathcal{F},h}$, there exist $g^1, \ldots, g^n \in \mathcal{F}$ so that $\mu_i$ is generated by executing $\pi_{g^i}$, for all $i \in [n]$.

By the definition of effective Bellman rank, this is equivalent to: $\sqrt{\sum_{i=1}^{t-1}(\langle \phi_h(f^t), \psi_h(g^i) \rangle)^2} \leq \epsilon$ and $|\langle \phi_h(f^t), \psi_h(g^t) \rangle| > \epsilon$ for all $t \in [n]$. For notational simplicity, define $x_i = \psi_h(g^i)$ and $\theta_i = \phi_h(f^i)$. Then

$$\begin{cases} \sum_{i=1}^{t-1}(x_i^\top \theta_t)^2 \leq \epsilon^2, & t \in [n] \\ |x_t^\top \theta_t| \geq \epsilon, & t \in [n]. \end{cases} \tag{24}$$

The remaining arguments follow the same as in the proof of Proposition 29 except that we replace $\epsilon$ by $\epsilon/\zeta$. $\qquad\square$

### H.4 Proof of Proposition 38

*Proof.* Note that the case $h = 1$ is trivial because each episode always starts from a fixed initial state independent of the policy. For any policy $\pi$, function $f \in \mathcal{F}$, and step $h \geq 2$

$$\begin{aligned} \mathcal{E}_{\mathrm{V}}(f, \pi, h) &= \mathbb{E}[f_h(o_h, a_h) - r_h(o_h, a_h) - f_{h+1}(o_{h+1}, a_{h+1}) \mid s_h \sim \pi, a_{h:h+1} \sim \pi_f] \\ &= \mathbb{E}[f_h(o_h, a_h) - r_h(o_h, a_h) - f_{h+1}(o_{h+1}, a_{h+1}) \mid (s_{h-1}, a_{h-1}) \sim \pi, a_{h:h+1} \sim \pi_f] \\ &= \sum_{s,a \in \mathcal{S}} \sum_{s' \in \mathcal{S}} \mathbb{P}^\pi(s_{h-1} = s, a_{h-1} = a) \cdot \langle \phi_{h-1}(s, a), \psi_{h-1}(s') \rangle_{\mathcal{H}} \cdot \mathcal{V}(s'), \end{aligned}$$

where

$$\mathcal{V}(s') = \mathbb{E}[f_h(o_h, a_h) - r_h(o_h, a_h) - f_{h+1}(o_{h+1}, a_{h+1}) \mid s_h = s', a_{h:h+1} \sim \pi_f].$$

As a result, we obtain

$$\begin{aligned} &\mathbb{E}[f_h(o_h, a_h) - r_h(o_h, a_h) - f_{h+1}(o_{h+1}, a_{h+1}) \mid s_h \sim \pi, a_{h:h+1} \sim \pi_f] \\ &= \left\langle \mathbb{E}_\pi[\phi_{h-1}(s_{h-1}, a_{h-1})], \sum_{s' \in \mathcal{S}} \psi_{h-1}(s')\mathcal{V}(s') \right\rangle_{\mathcal{H}}. \end{aligned}$$

Notice that the left hand side of the inner product only depends on $\pi$ while the right hand side only depends on $f$. Moreover, by the definition of kernel reactive POMDPs, the RHS has norm at most 2. Therefore, we conclude the proof by revoking Proposition 36 with $\zeta = 2$. $\qquad\square$

## I Discussions on $\mathcal{D}_{\mathcal{F}}$ versus $\mathcal{D}_\Delta$ in BE Dimension

In this paper, we have mainly focused on the BE dimension induced by two special distribution families: $(a)$ $\mathcal{D}_{\mathcal{F}}$ — the roll-in distributions produced by executing the greedy policies induced by the functions in $\mathcal{F}$, $(b)$ $\mathcal{D}_\Delta$ — the collection of all Dirac distributions. And we prove that both low $\dim_{\mathrm{BE}}(\mathcal{F}, \mathcal{D}_{\mathcal{F}}, \epsilon)$ and low $\dim_{\mathrm{BE}}(\mathcal{F}, \mathcal{D}_\Delta, \epsilon)$ can imply sample-efficient learning. As a result, it is natural to ask what is the relation between $\dim_{\mathrm{BE}}(\mathcal{F}, \mathcal{D}_{\mathcal{F}}, \epsilon)$ and $\dim_{\mathrm{BE}}(\mathcal{F}, \mathcal{D}_\Delta, \epsilon)$? Is it possible that one of them is always no larger than the other so that we only need to use the smaller one? We answer this question with the following proposition, showing that either of them can be arbitrarily larger than the other.

**Proposition 45.** *There exists absolute constant $c$ such that for any $m \in \mathbb{N}^+$,*

> *(a) there exist an MDP and a function class $\mathcal{F}$ satisfying for all $\epsilon \in (0, 1/2]$, $\dim_{\mathrm{BE}}(\mathcal{F}, \mathcal{D}_{\mathcal{F}}, \epsilon) \leq c$ while $\dim_{\mathrm{BE}}(\mathcal{F}, \mathcal{D}_\Delta, \epsilon) \geq m$.*

*(b) there exist an MDP and a function class $\mathcal{F}$ satisfying for all $\epsilon \in (0, 1/2]$, $\dim_{\mathrm{BE}}(\mathcal{F}, \mathcal{D}_\Delta, \epsilon) \leq c$ while $\dim_{\mathrm{BE}}(\mathcal{F}, \mathcal{D}_\mathcal{F}, \epsilon) \geq m$.*

*Proof.* We prove $(a)$ first. Consider the following contextual bandits problem $(H = 1)$.

- There are $m$ states $s_1, \ldots, s_m$ but the agent always starts at $s_1$. This means the agent can never visit other states because each episode contains only one step $(H = 1)$.

- There are two actions $a_1$ and $a_2$. The reward function is zero for any state-action pair.

- The function class $\mathcal{F}_1 = \{f_i(s, a) = \mathbf{1}(s = s_i) + \mathbf{1}(a = a_1) : i \in [m]\}$.

First of all, note in this setting $\mathcal{D}_\Delta$ is the collection of all Dirac distributions over $\mathcal{S} \times \mathcal{A}$, $\mathcal{D}_{\mathcal{F},1}$ is a singleton containing only $\delta_{(s_1, a_1)}$, and $(I - \mathcal{T}_1)\mathcal{F}$ is simply $\mathcal{F}_1$ because $H = 1$ and $r \equiv 0$. Since $\mathcal{D}_{\mathcal{F},1}$ has cardinality one, it follows directly from definition that $\dim_{\mathrm{BE}}(\mathcal{F}, \mathcal{D}_\Delta, \epsilon)$ is at most 1. Moreover, it is easy to verify that $(s_1, a_2), (s_2, a_2), \ldots, (s_m, a_m)$ is a 1-independent sequence with respect to $\mathcal{F}$ because we have $f_i(s_j, a_2) = \mathbf{1}(i = j)$ for all $i, j \in [m]$. As a result, we have $\dim_{\mathrm{BE}}(\mathcal{F}, \mathcal{D}_\Delta, \epsilon) \geq m$ for all $\epsilon \in (0, 1]$.

Now we come to the proof of $(b)$. Consider the following contextual bandits problem $(H = 1)$.

- There are 2 states $s_1$ and $s_2$. In each episode, the agent starts at $s_1$ or $s_2$ uniformly at random.

- There are $m$ actions $a_1, \ldots, a_m$. The reward function is zero for any state-action pair.

- The function class $\mathcal{F}_1 = \{f_i(s, a) = (2 \cdot \mathbf{1}(s = s_1) - 1) + 0.5 \cdot \mathbf{1}(a = a_i) : i \in [m]\}$.

First of all, note in this setting $(I - \mathcal{T}_1)\mathcal{F}$ is simply $\mathcal{F}_1$ and the roll-in distribution induced by the greedy policy of $f_i$ is the uniform distribution over $(s_1, a_i)$ and $(s_2, a_i)$, which we denote as $\mu_i$. It is easy to verify that $\mu_1, \ldots, \mu_m$ is a 0.5-independent sequence with respect to $\mathcal{F}$ because $\mathbb{E}_{(s,a) \sim \mu_i}[f_j(s, a)] = 0.5 \cdot \mathbf{1}(i = j)$. Therefore, for all $\epsilon \in (0, 0.5]$, $\dim_{\mathrm{BE}}(\mathcal{F}, \mathcal{D}_\mathcal{F}, \epsilon) \geq m$.

Next, we upper bound $\dim_{\mathrm{BE}}(\mathcal{F}, \mathcal{D}_\Delta, \epsilon)$ which is equivalent to $\dim_{\mathrm{DE}}(\mathcal{F}_1, \mathcal{D}_\Delta, \epsilon)$ in this problem. Assume $\dim_{\mathrm{DE}}(\mathcal{F}_1, \mathcal{D}_\Delta, \epsilon) = k$. Then there exist $g_1, \ldots, g_k \in \mathcal{F}_1$ and $w_1, \ldots, w_k \in \mathcal{S} \times \mathcal{A}$ such that for all $i \in [k]$, $\sqrt{\sum_{t=1}^{i-1}(g_i(w_i))^2} \leq \epsilon$ and $|g_i(w_i)| > \epsilon$. Note that we have $|f(s, a)| \in [0.5, 1.5]$ for all $(s, a, f) \in \mathcal{S} \times \mathcal{A} \times \mathcal{F}_1$. Therefore, if $\epsilon > 1.5$, then $k = 0$; if $\epsilon \leq 1.5$, then $k \leq 10$ because $0.5 \times \sqrt{k-1} \leq \sqrt{\sum_{t=1}^{k-1}(g_k(w_t))^2} \leq \epsilon \leq 1.5$. $\qquad\square$