# OpenReview forum: "Bellman Eluder Dimension: New Rich Classes of RL Problems, and Sample-Efficient Algorithms"
_NeurIPS.cc/2021/Conference — NeurIPS 2021 Spotlight_

### Official Review · Reviewer_bYxF · 2021-07-08

**Rating:** 7
**Confidence:** 4

**Summary:**

This paper introduces a new notion of dimension (i.e., the BE dimension) to measure the complexity of a function class when applied to RL problems. The BE dimension summarizes two previously prevailing notions: the Bellman rank and the Eluder dimension, and is also able to cover more tractable cases than before. An algorithm is proposed to be provably efficient with a low BE dimension which improves previous results based on low Bellman rank and low Eluder dimension.

**Limitations And Societal Impact:**

Yes.

**Main Review:**

I enjoy reading this paper. It is well-written. I especially like the authors' interpretations of each definition and the high-level explanation of the algorithm and the proof. I mainly have three points to comment on:

1. About the BE dimension

In my opinion, the construction of the BE dimension and the fact that it covers the Bellman rank and the Eluder dimension are not surprising. Please correct me if my understanding is wrong:

The main difference between the Bellman rank and the Eluder dimension is the underlying distribution (D_F or D_delta) and whether allowing non-linearity. BE dimension unifies them by introducing the freedom of selecting the distribution space. Once chosen proper distribution space and function space, BE dimension covers the Bellman rank due to a stronger constraint (see the below (b) ) and covers the eluder dimension with a smaller function space (F-TF \subset F-F).

What I find interesting/important is that BE covers more. In Proposition 13, the example illustrates two things:

a) F-TF can be much smaller than F-F and especially can exclude some pathological functions which can make the Eluder dimension explode. If the algorithm is of a Bellman-update fashion, analyzing F-TF is sufficient instead of going back to F-F.

b) BE dimension arouses stronger control than Bellman rank: Bellman rank is similar to \sqrt{\sum_{i=1}^{t-1} <q_t,w_i>^2 }\leq \sqrt(t-1)\epsilon and |<q_t, w_t>|>\epsilon (if we regard the inner product as projection), which relaxes the constraint in BE dimension to a scale of \sqrt{t}, therefore, allows more freedom. This might inspire future studies on how we should impose constraining conditions to determine a function.

Do you have any intuitive explanation on how BE covers more than bellman rank and eluder except for the freedom of selecting distributions?

2. About the benefit of BE.

What I care about most is: whether BE-based algorithms are more efficient than the other two; whether algorithms based on the other two can be transferred to BE-based versions. It seems that at least for Bellman-update type algorithms, BE dim is more satisfying than the other two. But for algorithms with on-policy critic fitting, e.g., ENIAC (https://arxiv.org/pdf/2103.11559.pdf), I wonder if BE can provide a better sample complexity than the Eluder dimension.

3. Strong condition on distributions.

Both D_F and D_\Delta contain possibly numerous distributions. I wonder if there is any idea on how to shrink the distribution set?

Minors:
1. At Line 221, an RL problem, not a.
2. For the Line 3 in Algo. 1, what does it mean by f(s_1, \pi_f(s_1))? Isn't f:= f_1 x f_2 ... x f_h? Or do you mean f_1(s_1, \pi_f(s_1))?
3. At line 217, should there be no comma in front of which?


**Time Spent Reviewing:**

10

---

> ### Author Response · Authors · 2021-08-10
> **Author Response**
>
> Thank you for your valuable time and suggestions. Please see our responses below:
>
>
> ${\bf Q}$.  Do you have any intuitive explanation on how BE covers more than Bellman rank and Eluder except for the freedom of selecting distributions?
>
> ${\bf A}$.
> Except for the freedom of selecting distributions, BE dimension is more general due to the following two reasons.
> - BE dimension measures the complexity of  *the Bellman residual* $\mathcal{F}-\mathcal{T}\mathcal{F}$ that could be much smaller than $\mathcal{F}-\mathcal{F}$ (see the example constructed in Proposition 13).
> -  BE dimension measures the *distributional Eluder dimension* of $\mathcal{F}-\mathcal{T}\mathcal{F}$, which is essential for handling nonlinear function class. In contrast, Bellman rank measures the linear dimension of $\mathcal{F}-\mathcal{T}\mathcal{F}$ which could be much larger than its (distributional) Eluder dimension.
>
>
>
> ${\bf Q}$.   For algorithms with on-policy critic fitting, whether BE can provide a better sample complexity than the Eluder dimension.
>
> ${\bf A}$.  At the current stage, it remains unclear to us how to directly use the techniques developed in this paper to design actor-critic type algorithms under BE dimension framework.
> However, we believe one promising approach is to construct confidence sets for the critic in a similar way to GOLF and update the actor with the most optimistic critic in the confidence set periodically, which is similar to the idea in [ZWB21] for the offline setting.  We agree it is an interesting future direction to explore.
>
>
>
>
> ${\bf Q}$.   Both $D_F$ and ${D}_{\Delta}$ contain possibly numerous distributions. I wonder if there is any idea on how to shrink the distribution set?
>
>
> ${\bf A}$. We believe the key quantity here  is the BE dimension instead of the distribution class. We remark that all the examples in this paper can be proved to have low BE dimension (which is sufficient to apply our theoretical results) despite their ${D}_F/{D}_\Delta$ being infinitely large. Although, we admit that in general providing an upper bound for BE dimension is a highly nontrivial step.
>
>
> ---
>
> $\bf{Reference}$
>
> [ZWB21] Provable Benefits of Actor-Critic Methods for Offline Reinforcement Learning. Andrea Zanette, Martin Wainwright, Emma Brunskill. ICML 2021 Workshop on Reinforcement Learning Theory.

---

> > ### Comment · Reviewer_bYxF · 2021-08-14
> > **Thanks for the answers.**
> >
> > The authors addressed all my questions. I vote for acceptance.

---

### Official Review · Reviewer_S8yr · 2021-07-12

**Rating:** 7
**Confidence:** 4

**Summary:**

This paper provides a solid and insightful analysis of the theoretical study of reinforcement learning. The reviewers think the contributions of the new definitions and techniques meet the standard of NeurIPS.

**Limitations And Societal Impact:**

yes

**Main Review:**

This paper proposes an extended version of the concept of the eluder dimension, which is originally used in bandit problems and later utilized in explorations in MDPs by several recent works. There are generally two branches of methods in this area, one models the value function, with the other modeling the transition kernel, by a general function set and characterizes the eluder dimension of such a function class. The new definition of the Bellman eluder dimension in this paper seems to connect these two cases as the definition involves both the value function class and the Bell operator. It is proved by the paper that the class of MDPs with structures defined by the Bellman eluder dimension is strictly more general than the previous classes, while the regret upper bound has the same form of $O(poly(H)\sqrt{Kd})$.


**Time Spent Reviewing:**

2

---

> ### Author Response · Authors · 2021-08-10
> **Author Response**
>
> Thank you for your valuable time and comments.

---

### Official Review · Reviewer_TbTF · 2021-07-19

**Rating:** 8
**Confidence:** 5

**Summary:**

This paper studies a new complexity measure, Bellman Eluder dimension, for provably efficient reinforcement learning. The authors propose algorithms that provably find near-optimal policies whose sample complexity can be upper bounded in terms of the Bellman Eluder dimension. The authors further show that for a large class of RL problems that admit sample-efficient algorithms, their Bellman Eluder dimension is bounded.

**Limitations And Societal Impact:**

See main review.

**Main Review:**

Recently, we have seen lots of progress on provably efficient RL. Existing work have already introduced a large set of structural assumptions on the underlying problem. Given this, an important open problem in this field is to unify existing assumptions and define new complexity measures that generalize existing assumptions. In this paper, the authors make important progress on this problem by defining the Bellman Eluder dimension which generalize a large set of existing assumptions.

Technically, Bellman Eluder dimension is defined as the (distributional version of) the eluder dimension of the Bellman residuals induced by the function class under consideration. Notable examples include function classes with bounded eluder dimension + Bellman completeness, and problems with low Bellman rank.

Overall, this is a great paper. It's pretty clear that it should be accepted. I will fight for its acceptance.

Comments

1. In Figure 1, why does LQR have bounded eluder dimension? It is known that LQRs satisfy Bellman completeness with linear function approximation. However, I don't see any reason why LQRs themselves have bounded eluder dimension.

2. Line 104. Generalized linear functions with unknown activations seem to be a bad example here. Latest version of [41] can indeed handle such functions (though [41] was updated after the NeurIPS submission due). Consider using other examples.

3. When discussing existing results, it could be helpful to mention the computation efficiency of the algorithms. E.g., although the algorithms in [7] and [17] require stronger representation conditions and achieve worst regret guarantee, they are computationally efficient assuming access to regression oracles. On the other hand, there is no existing computationally-efficient algorithm that achieves nearly optimal regret bound or only requires the weaker version of completeness.

4. It might be helpful to discuss problems that cannot be handle by the current framework. Notable examples include deterministic linear realizability (and its stochastic variants) [1-4].

[1] Efficient exploration and value function generalization in deterministic systems

[2] Provably Efficient Q-learning with Function Approximation via Distribution Shift Error Checking Oracle

[3] Agnostic Q-learning with Function Approximation in Deterministic Systems: Near-Optimal Bounds on Approximation Error and Sample Complexity

[4] On Query-efficient Planning in MDPs under Linear Realizability of the Optimal State-value Function



**Time Spent Reviewing:**

10

---

> ### Author Response · Authors · 2021-08-10
> **Author Response**
>
> Thank you for your valuable time and suggestions. Please see our responses below:
>
>
>
> ${\bf Q}$.  In Figure 1, why does LQR have bounded Eluder dimension? It is known that LQRs satisfy Bellman completeness with linear function approximation. However, I don't see any reason why LQRs themselves have bounded Eluder dimension.
>
>
> ${\bf A}$. As noted by the reviewer, LQRs satisfy linear function approximation with completeness.
> It is also known [e.g., RR13] linear function class has low Eluder dimension. Therefore, LQRs satisfy low Eluder dimension with completeness, which implies low BE dimension according to Proposition 12. We will clarify this more in our revisions.
>
>
> ${\bf Q}$.    Line 104. Generalized linear functions with unknown activations seem to be a bad example here. Latest version of [41] can indeed handle such functions (though [41] was updated after the NeurIPS submission due). Consider using other examples.
>
>
> ${\bf A}$. Thanks for pointing out this recent update of [41], we will modify our statements accordingly. In order to handle generalized linear function approximation, [41] requires a modified version of bilinear class and provides suboptimal sample complexity guarantee $\tilde{\mathcal{O}}({d^5}/{\epsilon^4})$. In contrast, the BE framework naturally incorporates the setting of generalized linear function approximation without any modification, and provides  sharper regret guarantee  $\tilde{\mathcal{O}}(d\sqrt{T})$ (which implies $\tilde{\mathcal{O}}({d^2}/{\epsilon^2})$ sample complexity guarantee). Finally, although [41] can deal with generalized linear function approximation, it does not capture  the class of problems with low Eluder dimension [e.g., WSY20] while this work can.
>
>
>
> ${\bf Q}$.  Discuss the computational efficiency of related works and problems that can not be handled by the current framework.
>
> ${\bf A}$.
> Thank you for the helpful suggestions.
> We agree that the algorithms proposed in our work as well as Olive [JKA+17] and Elenar [ZLKB20] are computationally inefficient even when reduced to the tabular setting. We also agree that several existing works [e.g., 7,17] can be computationally efficient given suitable regression oracles at the cost of stronger representation conditions and worse regret guarantees. We will clarify this point in the revision.
> We will also add discussions on problems that are not covered by the current framework, e.g., the deterministic linear realizability setting mentioned by the reviewer.
>
>
>
>
> ---
> ${\bf Reference}$
>
> [RR13]  Eluder Dimension and the Sample Complexity of Optimistic Exploration. Daniel Russo, Benjamin Van Roy. NIPS, 2013.
>
> [JKA+17] Contextual Decision Processes with low Bellman rank are PAC-Learnable. Nan Jiang, Akshay Krishnamurthy, Alekh Agarwal, John Langford, Robert E. Schapire. ICML, 2017.
>
>
> [ZLKB20] Learning Near Optimal Policies with Low Inherent Bellman Error. Andrea Zanette, Alessandro Lazaric, Mykel Kochenderfer, Emma Brunskill. ICML, 2020.
>
>
> [WSY20] Reinforcement Learning with General Value Function Approximation: Provably Efficient Approach via Bounded Eluder Dimension.
> Ruosong Wang, Ruslan Salakhutdinov, Lin F. Yang.
> NeurIPS 2020.

---

### Official Review · Reviewer_T8on · 2021-07-20

**Rating:** 7
**Confidence:** 4

**Summary:**

The paper provides a new set of assumptions under which,  given a realizable Q-function / V-function class, one can efficiently find an almost optimal policy. They define the notion of bellman eluder (BE) dimension and provide regret bounds for RL problems for which the BE dimension is bounded. In particular, the regret bounds do not depend on the size of the state space |S|, and only scale with BE dimension, H and log(|F|), where F denotes the given realizable value function class. Their algorithm is based on optimism with local confidence intervals.

They also show that almost all the RL settings where we know how to get a sample efficient algorithm have small BE dimensions. In addition to this, they also provide an example setting (kernel reactive POMDPs) where BE dimension is small and hence their algorithm is sample efficient, but we did not know any algorithm for this setting before the paper.

**Main Review:**

I support accepting the paper. However, I have a few concerns and would be open to increase my score depending on the feedback:

(1. ) Can we test Assumption 2 ? Given a realizable value function class, how can I test whether Assumption 2 holds and if I can run GOLF?

(2.) What happens when F is not realizable. Suppose it is \epsilon-approximately realizable. What kind of guarantees can we hope for in this case?

(3.) Can GOLF be implemented efficiently for any RL setting? What about tabular RL?

(4.)  Is there any RL problem setting where we know how to solve it as a stand alone problem, but the BE dimension is not small?

(5.) From what I understand, Kernel reactive POMDPs is the only problem class which was not captured by bi-linear classes framework of [41]. All other problems are captured by the framework of [41] as well. Do you / they have a proof that Kernel reactive POMDPs are not captured by the framework of [41]? I worry that the framework of [41] is equally general. In that case, the framework of BE dimension seems to add nothing new.

(6.) Can we compare the BE dimension with the distribution class D_{F} and D_{\Delta}. Is one always larger than the other?

The paper is very well written. Minor typo: Line 338: discusses -> discuss.


**Time Spent Reviewing:**

8-10 hours

---

> ### Author Response · Authors · 2021-08-10
> **Author Response**
>
> Thank you for your valuable time and suggestions. Please see our responses below:
>
>
> ${\bf Q}$. Can we test Assumption 2? Given a realizable value function class, how can I test whether Assumption 2 holds and if I can run GOLF?
>
> ${\bf A}$. If we have prior knowledge about the structure of the MDPs (e.g., linear or generalized linear or LQR), then we can verify realizability and completeness. Otherwise, it remains unclear  how to verify both conditions for arbitrary unknown MDPs. This challenge for verification persists in most of the prior works [e.g., JKA+17, WSY20, DKL+21] that study general function approximation. We believe it is an important open question worth further investigation.
>
>
> ${\bf Q}$. What happens when $\mathcal{F}$ is not realizable. Suppose it is $\delta$-approximately realizable. What kind of guarantees can we hope for in this case?
>
> ${\bf A}$. We can readily extend the current guarantees to the setting where realizability and completeness hold with an approximation error $\delta$. In that case, we only need to  increase the confidence parameter $\beta$ by $K\delta^2$ and slightly modify the concentration analysis (Lemma 38, 39). The remaining analysis will be the same. The final regret bound in Theorem 15 will have an additional term $KH\delta\sqrt{d_{\rm BE}}$.
>
>
>
> ${\bf Q}$. Can GOLF be implemented efficiently for any RL setting? What about tabular RL?
>
> ${\bf A}$. Unfortunately, the original GOLF algorithm cannot be efficiently implemented in the tabular setting, as it requires to solve a jointly nonconvex optimization problem (step 3 in Algorithm 1) which may cost exponential time in the worst case.
> Nevertheless, if we allow certain relaxations of the confidence set, then GOLF can be relaxed to some variant of VI-UCB [AOM17], which can be efficiently implemented by greedy dynamic programming.
>
>
>
> ${\bf Q}$.  Is there any RL problem setting where we know how to solve it as a stand alone problem, but the BE dimension is not small?
>
> ${\bf A}$. As mentioned by Reviewer TbTF, the current framework can not incorporate deterministic linear realizable settings without completeness [WR13]. Besides, since our work focuses on value-based function approximation, it can not address examples that are naturally more suitable for model-based function approximation such as factored MDPs.
>
>
>
> ${\bf Q}$.  From what I understand, Kernel reactive POMDPs is the only problem class which was not captured by bilinear classes framework of [41]. All other problems are captured by the framework of [41] as well. Do you/they have a proof that Kernel reactive POMDPs are not captured by the framework of [41]? I worry that the framework of [41] is equally general. In that case, the framework of BE dimension seems to add nothing new.
>
> ${\bf A}$.
> We thank the reviewer for raising the connections to this nice related work [41]. We would like to comment that (1) [41] and this work are concurrent works, and (2) [41] does not cover our work in the following perspectives.
>
> - Kernel reactive POMDPs are also captured by [41]. However, [41] does not capture the class of problems with low Eluder dimension [e.g., WSY20] while this work can.
> - For problems captured by both frameworks, our framework can provide sharper guarantees in certain settings. For example, in the setting of linear function function with completeness, we recover the optimal regret guarantee $\tilde{\mathcal{O}}(d\sqrt{T})$ (which implies $\tilde{\mathcal{O}}({d^2}/{\epsilon^2})$ sample complexity guarantee). In contrast, [41] only provides suboptimal $\tilde{\mathcal{O}}({d^3}/{\epsilon^2})$ sample complexity guarantee. Similarly in the setting of generalized linear function function with completeness, we still have $\tilde{\mathcal{O}}(d\sqrt{T})$ regret guarantee while [41] provides suboptimal $\tilde{\mathcal{O}}({d^5}/{\epsilon^4})$ sample complexity guarantee.
> - This paper also introduces a new simple algorithm GOLF which can be viewed as an optimistic version of the classical algorithm---fitted Q-iteration. This is not included in [41].
>
>
> ${\bf Q}$.  Can we compare the BE dimension with the distribution class $D_F$ and $D_{\Delta}$. Is one always larger than the other?
>
> ${\bf A}$. Either of them can be arbitrarily  larger than the other. Please refer to Appendix I for details.
>
>
> ---
> $\bf{Reference}$
>
> [WR13] Efficient Exploration and Value Function Generalization in Deterministic Systems. Zheng Wen, Benjamin Van Roy. NIPS, 2013.
>
>
> [AOM17] Minimax Regret Bounds for Reinforcement Learning. Mohammad Gheshlaghi Azar, Ian Osband, Rémi Munos. ICML, 2017.
>
>
>
> [JKA+17] Contextual Decision Processes with low Bellman rank are PAC-Learnable. Nan Jiang, Akshay Krishnamurthy, Alekh Agarwal, John Langford, Robert E. Schapire. ICML, 2017.
>
> [WSY20] Reinforcement Learning with General Value Function Approximation: Provably Efficient Approach via Bounded Eluder Dimension.
> Ruosong Wang, Ruslan Salakhutdinov, Lin F. Yang.
> NeurIPS 2020.
>
> [DKL+20] Bilinear Classes: A Structural Framework for Provable Generalization in RL. Simon S. Du, Sham M. Kakade, Jason D. Lee, Shachar Lovett, Gaurav Mahajan, Wen Sun, Ruosong Wang. ICML 2021.

---

> > ### Comment · Reviewer_T8on · 2021-09-13
> > **Final comments**
> >
> > I have read other reviews and the author feedback and would like to keep my review unchanged. I continue to support acceptance for this paper!

---

### Decision · Program_Chairs · 2021-09-27

**Decision:**

Accept (Spotlight)

**Comment:**

The paper introduces a new structural assumption for provably efficient online reinforcement learning. The authors provided a characterization of the new Bellman Eluder Dimension, showing that it generalizes a large part of existing structural assumptions while preserving efficient learning. Overall, the paper is novel, well written and technically solid. I think it should be accepted.